# Strain-graded quantum dots with spectrally pure, stable and polarized emission

Dongju Jung [1,6], Jeong Woo Park [1,6], Sejong Min [1], Hak June Lee[1], Jin Su Park[1], Gui-Min Kim[2], Doyoon Shin [1], Seongbin Im [1], Jaemin Lim [1], Ka Hyung Kim[1], Jong Ah Chae[3], Doh C. Lee [2], Raphaël Pugin[4], Xavier Bulliard [4] ✉, Euyheon Hwang [1], Ji-Sang Park [1] ✉, Young-Shin Park [5] ✉ & Wan Ki Bae [1,3] ✉

Structural deformation modifies the bandgap, exciton fine structure and phonon energy of semiconductors, providing an additional knob to control their optical properties. The impact can be exploited in colloidal semiconductor quantum dots (QDs), wherein structural stresses can be imposed in three dimensions while defect formation is suppressed by controlling surface growth kinetics. Yet, the control over the structural deformation of QDs free from optically active defects has not been reached. Here, we demonstrate strain-graded CdSe-ZnSe core-shell QDs with compositionally abrupt interface by the coherent pseudomorphic heteroepitaxy. Resulting QDs tolerate mutual elastic deformation of varying magnitudes at the interface with high structural fidelity, allowing for spectrally stable and pure emission of photons at accelerated rates with near unity luminescence efficiency. We capitalize on the asymmetric strain effect together with the quantum confinement effect to expand emission envelope of QDs spanning the entire visible region and exemplify their use in photonic applications.

Colloidal quantum dots (QDs) are free-standing nano-emitters that radiate photons of size-dependent adjustable wavelength with a narrow emission linewidth[1,2]. Core-shell heterostructuring in these nanometer scale semiconductors allows one to tailor their photophysical characteristics[3–10]. Heteroepitaxy involves structural deformation of both core and shell materials in a way that the structural stress at the interface is efficiently alleviated. The structural deformation of core materials often appears to be enough to modify the electronic structure and photophysical characteristics[11–13]. Specifically, asymmetric lattice strain imposed on QDs lifts the dark exciton states above the bright excitons and splits the bright excitons into two distinct states[12,14,15], accompanying the accelerated radiative decay, narrowed spectral

linewidths and the reduced optical gain threshold which are crucial for QD applications in displays[16–18], lasers[19–21] and single-photon sources[22].

The degeneracy lift in colloidal QDs was first reported from a spherical core-asymmetrically grown shell geometry[19] and has been found even in a conventional spherical core-shell geometry[16–18,21–24] Despite progress, the relation between the structural factors and photophysical/electronic characteristics of results QDs remains unknown. The structural complexity of QD heterostructures, which includes an asymmetric geometry[19] of shell or variations in composition[16–18,20–23] or crystal structure[24] of shell along with misfit defects of near the interface, is the main culprit that impedes systematic investigation on the degeneracy lift in QDs. This calls for a

[1]SKKU Advanced Institute of Nanotechnology (SAINT), Sungkyunkwan University (SKKU), Suwon 16419, Republic of Korea. [2]Department of Chemical and Biomolecular Engineering, KAIST Institute for the Nanocentury, Korea Advanced Institute of Science and Technology (KAIST), Daejeon 34141, Republic of Korea. [3]Department of Display Engineering, Sungkyunkwan University (SKKU), Suwon 16419, Republic of Korea. [4]Centre Suisse d'Electronique et de Microtechnique (CSEM SA), CH-2002 Neuchatel, Switzerland. [5]Chemistry Division, Los Alamos National Laboratory, Los Alamos, NM 87545, USA. [6]These authors contributed equally: Dongju Jung, Jeong Woo Park. ✉e-mail: xavier.bulliard@csem.ch; jisangpark@skku.edu; ypark19@lanl.gov; wkbae@skku.edu

materials platform that reaches degeneracy lift in core-shell heterostructure of a tailored geometry without misfit defect formation.

In the present study, we devise a coherent pseudomorphic growth technique for ZnSe shell over CdSe core to reach degeneracy lift in a conventional core-shell geometry. Resulting CdSe-ZnSe QDs bear compositionally abrupt but strain-graded core-shell interface, in which the compressive strain imposed in CdSe is greatly intensified while the formation of misfit defects is effectively suppressed. We perform comprehensive study across structural analysis, ensemble and single-dot spectroscopic analyses and computational calculations on the strain-graded QDs of variable core-shell dimensions, and relate the structural factors and their photophysical properties. Finally, we reveal the potential of coherent pseudomorphic growth of centered core-shell geometry to expand the emission energy envelope of strain-graded QDs, covering a wider spectral range and opening new avenues for applications in photonics.

## Results

### Strain-graded QDs with compositionally abrupt interface

Asymmetry in geometry[19] or crystal structure[22] of shell layers is the impetus for asymmetric strain in the core. The former deploys facet dependent anisotropic growth of shell epilayers on the core, while the latter employs the different compressive stress given by the reduced crystal symmetry of materials. The geometric approach is indeed a double-edged sword to the optical performance of resulting heterostructured QDs. It is well known that the thick shell side accumulates structural stress at the core-shell interface to create misfit defects, whereas the thin shell side is ineffective to confine charge carriers, both of which are responsible for inefficiency and instability of QD's emission characteristics. By contrast, the other approach is applicable to attain asymmetrically strained QDs in a conventional core-shell geometry. This scheme is highly effective when the formation of strain-relieving misfit defects is suppressed, which otherwise lessens the impact of imposed strain and deteriorates the optical performance of QDs. Core-compositionally graded shell heterostructure, as known as cg-QD[22,25], has been elaborated to meet the criteria, but it inevitably accompanies the reduction of the magnitude of the compressive stress onto the core as well as the spectral broadening originating from the compositional or structural inhomogeneity of the interface layer.

To exploit the impact of compressive strain, we grow asymmetrically strained wurtzite (wz) CdSe-wz ZnSe QDs in an centered core-shell geometry with compositionally abrupt interface (Fig. 1a, b). Here, the key to enhance the strain imposed in CdSe core is to suppress the formation of unintentional interfacial alloys made of mixed cations, i.e., Cd$_{1-X}$Zn$_X$Se, during the ZnSe shell growth. Our Density functional theory (DFT) calculation predicts that the polar surface planes of cation (Cd)-rich CdSe cores stabilize their surfaces by leaving cation vacancies (Supplementary Fig. 1 and Note 1), which are fated to be filled by Zn upon the subsequent ZnSe growth to yield Cd$_{1-X}$Zn$_X$Se alloy interfacial layers. To avoid unintentional alloying at the interface, we start from CdSe core terminated by anions (Se), whose vacancies at the surfaces are not relevant to the formation of cation mixed alloy layers during the ZnSe epitaxial growth (Supplementary Figs. 2, 3). The layer-by-layer growth of ZnSe shell on CdSe core at an elevated reaction temperature ($T \geq 340\,^{\circ}$C) is deployed for the thermodynamic ZnSe growth (see Methods). Specifically, the growth rate of ZnSe epilayer is engineered by means of cation-to-ligand stoichiometric control of Zn precursor and its feed rate to accomplish core-shell heteroepitaxy free from optically active defects (Supplementary Fig. 4).

Resulting CdSe-ZnSe QDs show near unity photoluminescence quantum yields (PL QYs) and single exponential decay dynamics even for ZnSe shell thicknesses exceeding the critical shell thickness[26] (ca. 2.33 nm, the limit thickness for flat ZnSe epilayers that can be grown on CdSe bulk film without the formation of misfit defects, Supplementary Fig. 5 and Note 2), implying that CdSe core and ZnSe shell are mutually

strained at the interface to alleviate the lattice mismatch (i.e., −5.48% (−7.05%) along [0002] ([1000]) axis in bulk CdSe versus ZnSe[27]) and suppress the misfit defect formation. These QDs bear compositional discontinuity at the interface but strain gradient across the core-shell interface (hereinafter, referred to as strain-graded (sg)-QDs) to build a smooth potential profile along the radial direction (Fig. 1a). The sg-QDs have a centered core-shell geometry, which make them uniformly strained and effectively confine charge carriers (Fig. 1b). However, compressive strain in CdSe core develops differently depending on the crystal axes. Specifically, in CdSe (radius, $r = 2.5$ nm)-ZnSe (shell thickness, $H = 5.0$ nm) heterostructure, the mean compressive strain ($\beta$, $\Delta d/d \times 100$ (%)) of entire CdSe core regime along [0002] and [11$\bar{2}$0] are measured to be −3.08% and −4.33%, respectively (Fig. 1c–e and Supplementary Fig. 6). This asymmetric strain between the basal direction, [11$\bar{2}$0], and its orthogonal direction, [0002] is in good agreement with our DFT calculation results (Supplementary Fig. 7 and Note 3).

As reported in previous studies of asymmetric QDs[16–19,21–23], our sg-QDs features the degeneracy lift of exciton states, as signified with the energy split of the first exciton states in absorption and PL spectra and accelerated radiative recombination rates (Fig. 1f–i). However, due to maximized asymmetric strain in sg-QDs with centered CdSe core, the energy splitting is as large as 50 meV and the exciton decay is close to purely radiative with $\tau_X = 10$ ns[12]. Notably, our sg-QDs show stable PL emission with a narrow spectral linewidth (17.7 meV) and suppressed spectral diffusion (std = 0.27 meV) in an individual dot level (Fig. 1j) and small dot-to-dot variations of emission characteristics (Fig. 1k), accounting for the record-narrow ensemble PL spectra among colloidal core-shell nano-emitters (Supplementary Fig. 8). These characteristics corroborate the augmented impact of compressive strain and reduced compositional and/or structural inhomogeneity in given CdSe-ZnSe sg-QDs with compositionally abrupt interface.

## Photophysical characteristics of ensemble sg-QDs

The thermodynamic ZnSe growth provides us with an effective approach to induce the asymmetric strain in CdSe having dimensional variations of CdSe core radii (2.0 nm $\leq r \leq 4.0$ nm) and ZnSe shell thicknesses (1.0 nm $\leq r \leq 5.0$ nm), enabling us to explore the relationship between CdSe-ZnSe QD structure and their electronic/optical properties (Fig. 2). Figure 2a, b show absorption and PL spectra for sg-QDs with $r = 2.5$ nm and varying $H$. Immediately upon the ZnSe epilayer growth, the exciton energy increases and splits into two distinct peaks that are the heavy hole exciton state (1S$_e$ → 1S$_{HH}$ transition, referred to as 1S$_{HX}$) and the light hole exciton state (1S$_e$ → 1S$_{LH}$ transition, referred to as 1S$_{LX}$) (Fig. 2a, b)[12,28,29]. Recovery of exciton energy to the original position upon the chemical etching (Supplementary Fig. 9 and Method) suggests that the increase in exciton energy is indeed attributed to the compressive strain imposed in CdSe core by ZnSe shell rather than Cd-Zn inter-diffusion at the interface[30,31].

Considering the linear relationship between the compressive strain and the expansion of the bulk bandgap of CdSe[32], the magnitude of the effective compressive strain in CdSe core ($\beta^*$) can be estimated from the changes in optical properties of CdSe-ZnSe sg-QDs with the effective mass approximation (Fig. 2c, Supplementary Table 1 and Note 4). In chosen dimensional variations, the magnitude of compressive strain in CdSe core increases with thicker ZnSe epilayers and the impact is more pronounced for smaller CdSe core (Fig. 2c). Interestingly, the magnitude of HH-LH energy splitting in the exciton transitions, a signature of asymmetric strain in CdSe core[22], is not related to the magnitude of compressive strain, but associated with asymmetric compressive stress along axes at a given crystal structure together with morphological asymmetry of grown ZnSe shells. This would explain the tendency – the magnitude of energy splitting increases along the growth of ZnSe epilayers to reach the peak ($\Delta E_{split} = 55 - 60$ meV at $H = 2.0$–$3.0$ nm) and decreases down to

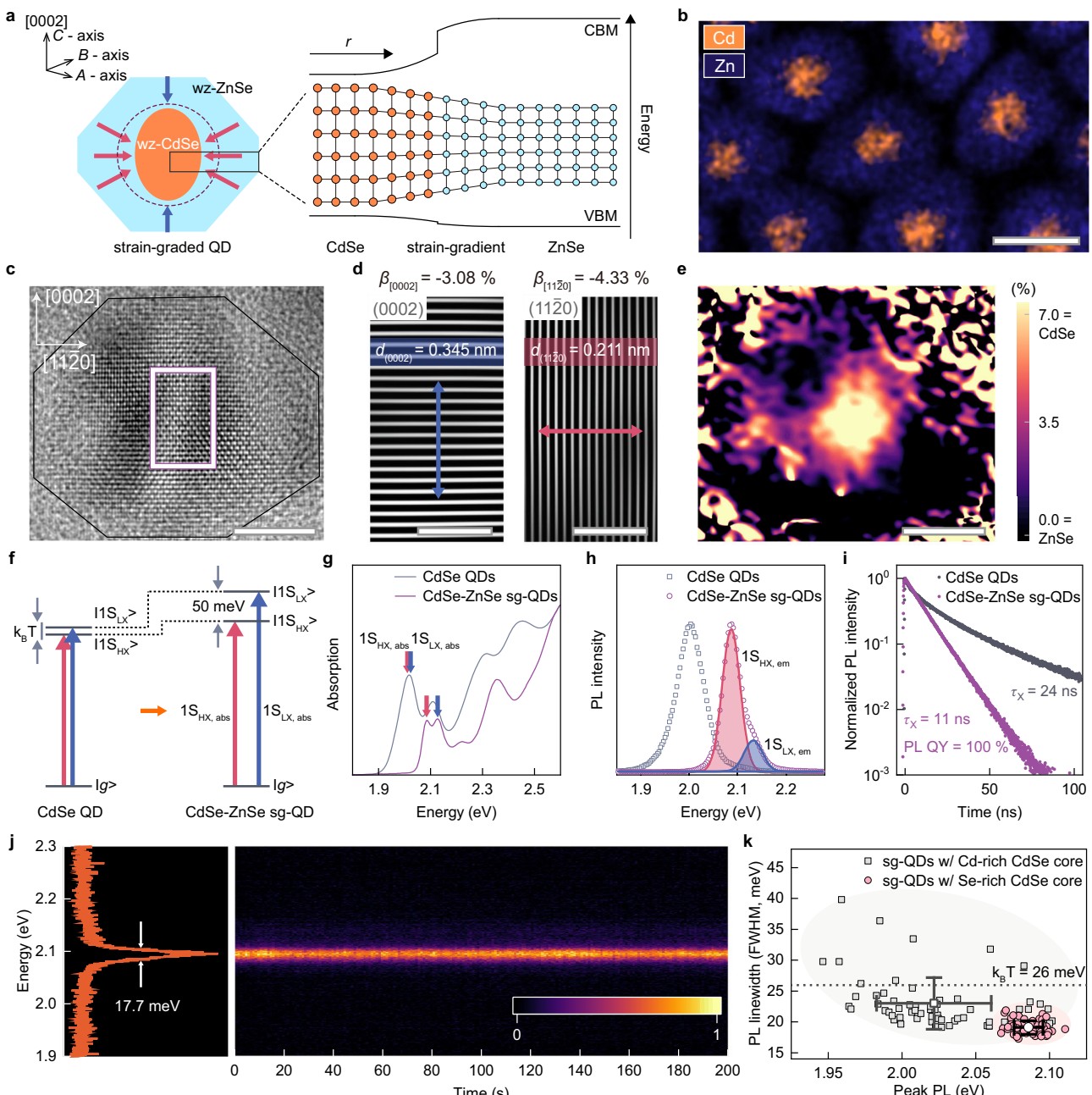

**Fig. 1 | Strain-graded CdSe-ZnSe core-shell QDs with compositionally abrupt interface. a** Schematic illustration of strain-graded (sg) wurtzite (wz) CdSe-ZnSe centered core-shell QDs with compositionally abrupt interface. The spherical CdSe core, under greater compression perpendicular to the $C$ axis (red) than along the $C$ axis (blue), becomes prolate spheroid-shaped. Near the interface, CdSe and ZnSe are mutually strained, creating smooth potential gradients in conduction and valence band edges. **b** EDS elemental mapping of Cd (red) and Zn (green) and **c**, HR-TEM image of CdSe ($r = 2.5$ nm)-ZnSe ($H = 5.0$ nm) sg-QDs. **d** Inverse FFT images of CdSe core (white square in (**c**)) showing the contraction of CdSe along [0002] (blue) and [11$\bar{2}$0] (red). Mean d-spacing and magnitude of compressive strain ($\beta$) along these directions are noted. **e** Lattice constant map obtained by geometric phase analysis. Scale bars in (**b**–**e**) are 10 nm, 5 nm, 1 nm, 1 nm and 5 nm, respectively. **f** Energy level diagram showing the changes in optically active exciton states

upon the asymmetric compressive strain. $|1S_{HX}>$, $|1S_{LX}>$ and $|g>$ represent the state of heavy hole exciton, light hole exciton and ground, respectively. **g** Absorption spectra, **h** PL spectra and (**I**), PL decay dynamics of CdSe QDs ($r = 2.5$ nm) (grey) and CdSe ($r = 2.5$ nm)-ZnSe ($H = 5.0$ nm) sg-QDs (purple). Red and blue arrows in (**g**) indicate optical transitions from $|g>$ to $|1S_{HX}>$ and $|1S_{LX}>$, respectively. PL spectrum of sg-QDs in (**h**) is fitted with double gaussian curves, $1S_{HX,em}$ ($|1S_{HX}> \rightarrow |g>$, red line with shading) and $1S_{LX,em}$ ($|1S_{LX}> \rightarrow |g>$, line with blue shading). Single exciton decay times ($\tau_X$) are noted in (**I**). **j** PL spectrum of an individual sg-QD (left) and a 2D contour plot showing PL spectra of 200 sequential frames (1 s per frame) (right) and (**k**), PL energy and linewidth of individual sg-QDs with Se-rich and Cd-rich CdSe core for CdSe ($r = 2.5$ nm)-ZnSe ($H = 5.0$ nm) sg-QDs (symbols: average, error bars: standard deviation, shaded region: range guidance) More structural information is provided in Supplementary Fig. 6.

50 meV when bulk like ZnSe shell ($H \geq 4.0$ nm) is grown (Fig. 2d). We attribute this intriguing behavior to the nonlinear change of the asymmetric strain for the thickness of the ZnSe shell (Supplementary Fig. 7 and Note 3), although the exact thickness might be different due to the simplified geometry in our calculation.

The enlargement of compressive strain in CdSe core by thicker ZnSe shells comes with accelerated radiative recombination rates (Figs. 1i, 2e), demonstrating the progression of the lift of dark exciton states apart from the bright ones with an increased compressive strain[22,33]. The shortening of radiative recombination time appears

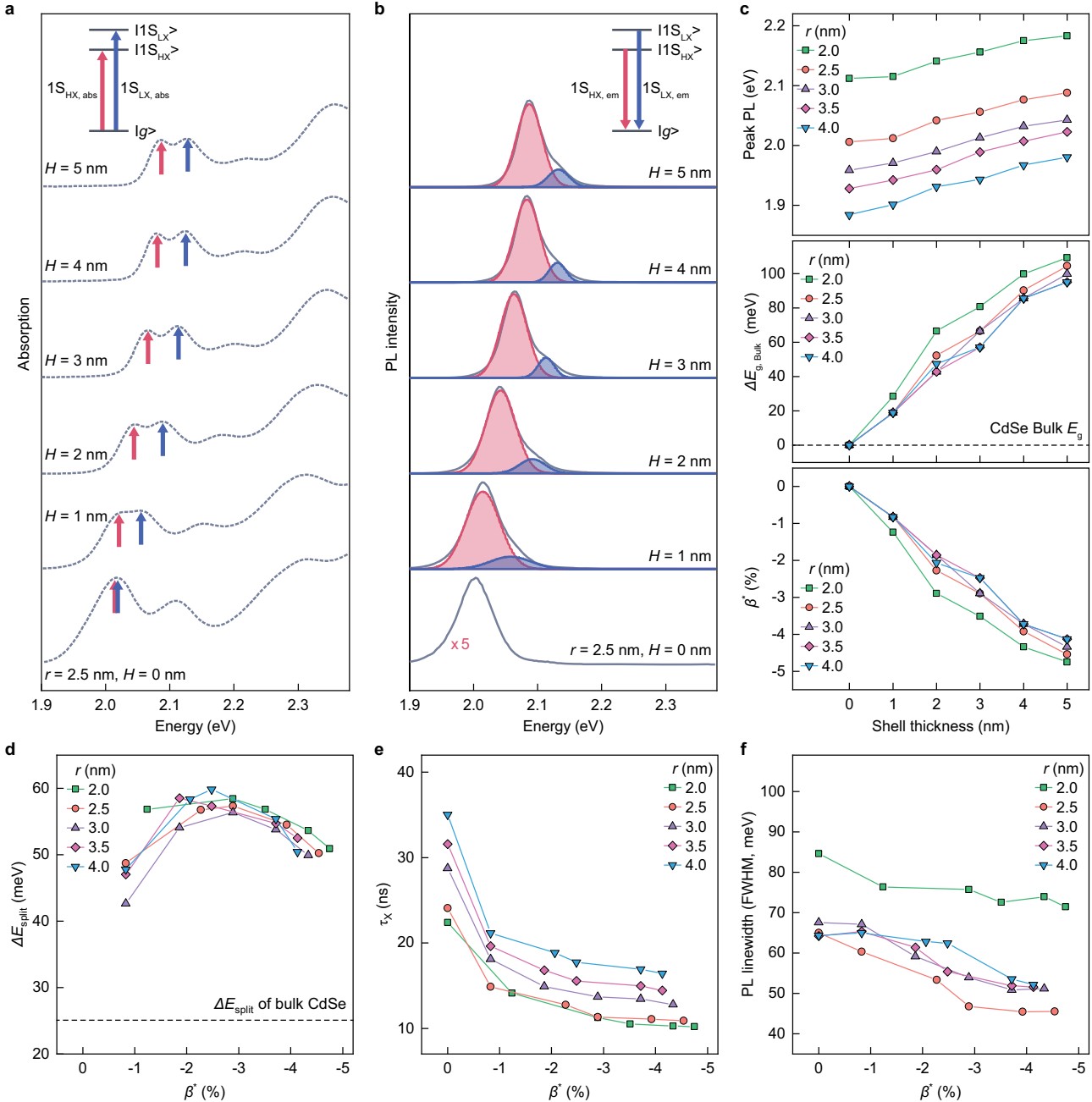

**Fig. 2 | Photophysical characteristics of CdSe-ZnSe sg-QD ensembles with varying core-shell geometries. a** Absorption spectra and (**b**), PL spectra of CdSe ($r = 2.5$ nm)-ZnSe sg-QDs with varying shell thicknesses (0 nm $\leq H \leq$ 5.0 nm). Arrows in (**a**) and shading in (**b**) indicate optical transitions between $|1S_{HX}\rangle$ and $|g\rangle$ (red) and $|1S_{LX}\rangle$ and $|g\rangle$ (blue). **c** Peak PL energy of CdSe-ZnSe sg-QDs with varying core radii (2.0 nm $\leq r \leq$ 4.0 nm) and shell thicknesses (0 nm $\leq H \leq$ 5.0 nm) (upper panel), and estimated changes in the bulk band gap ($\Delta E_{g,Bulk}$) of CdSe (middle panel) and

the effective compressive strain ($\beta^*$) (lower panel) developed in CdSe core upon ZnSe shell growth. **d** Heavy hole-light hole energy split ($\Delta E_{split}$), **e** radiative recombination time for single exciton ($\tau_X$) and (**f**), ensemble emission linewidth of CdSe-ZnSe sg-QDs with varying core radii (2.0 nm $\leq r \leq$ 4.0 nm) as a function of the effective compressive strain imposed by ZnSe of varying shell thicknesses (0 nm $\leq H \leq$ 5.0 nm). More sample information is provided in Supplementary Fig. 12.

regardless of the core dimension, and it converges to *ca.* 45% of the characteristics time of CdSe core, close to the purely radiative recombination time[12]. In addition, PL spectrum from $|1S_{HX}\rangle$ becomes narrower with thicker ZnSe shells (Fig. 2f, see also Fig. 3). We attribute the reasons for the PL spectral linewidth narrowing to the suppressed exciton-phonon coupling with increased optical phonon energies ("phonon hardening") in the compressively strained CdSe (Supplementary Fig. 10, Table 2 and Note 5)[34] along with the suppressed spectral diffusion with reduced coupling of charge carriers to the environment by thick energy barrier layers (Supplementary Fig. 11)[22].

## Emission characteristics of individual sg-QDs

The type I band alignment with the smooth potential profile across CdSe core and ZnSe shell in sg-QDs facilitates funneling of charge carriers[35] into CdSe core, as reflected by the suppression of fluorescence intermittency (blinking) under the continued photo-irradiation (Fig. 3a). In addition, the smooth potential profile at the interface aids to suppress non-radiative Auger recombination processes[36] (Fig. 3b and Supplementary Fig. 13). These effects are orchestrated to allow for the stable PL emission of our sg-QDs under varying pump intensities (Supplementary Fig. 14). The direct pseudomorphic growth of ZnSe

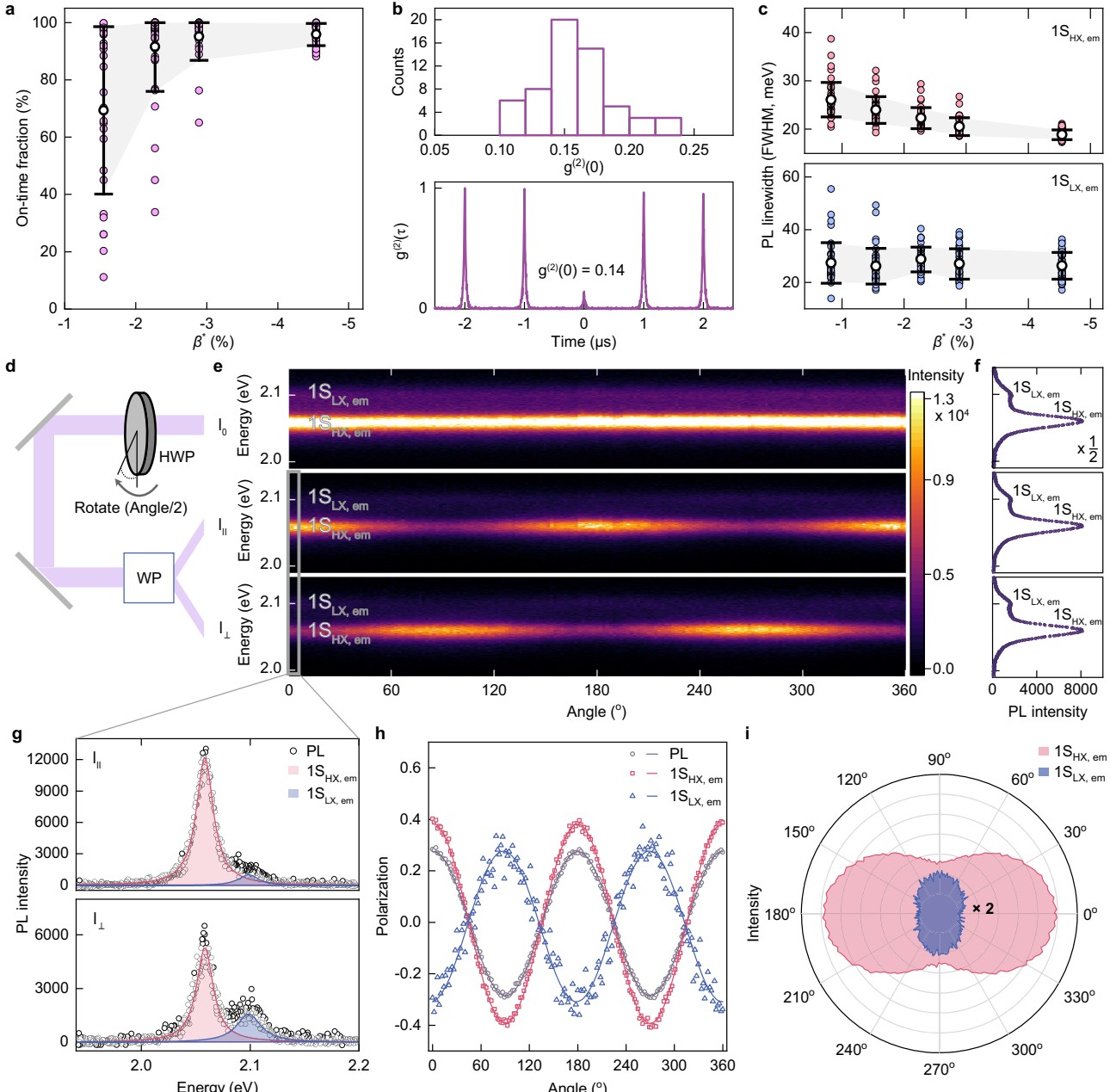

**Fig. 3 | Emission characteristics of individual sg-QDs. a** On-time fractions, (**b**), histogram of $g^{(2)}(0)$ (upper panel) derived from the second-order correlation function, $g^{(2)}(\tau)$, and a representative graph (lower panel), and (**c**), emission linewidths of PL spectrum components, $1S_{HX,em}$ (upper panel) and $1S_{LX,em}$ (lower panel), of individual CdSe ($r = 2.5$ nm)-ZnSe sg-QDs as a function of the effective compressive strain imposed by ZnSe of varying shell thicknesses (0 nm ≤ $H$ ≤ 5.0 nm). The $g^{(2)}(\tau)$ in (**b**) is obtained from a CdSe ($r = 2.5$ nm)-ZnSe ($H = 5.0$ nm) sg-QD and the mean value of $g^{(2)}(0)$ is 0.16. In (**a**, **c**), symbols are averages, error bars are standard deviations, and shaded regions are connecting the error bars. **d** Schematic illustration of the optical measurement setup. The emission from a single sg-QD, $I_0$, passes through a half-wave plate (HWP), which rotates the polarization direction of the light, and a Wollaston prism (WP), which splits the emission into two linearly polarized beams with orthogonal polarization ($I_\parallel$ and $I_\perp$). **e** 2D contour plots showing PL spectra ($I_0$, $I_\parallel$ and $I_\perp$ from the top) of an individual sg-QD at varying rotation angles from 0 to 360 degrees (100 ms per each degree) and (**f**), averaged PL spectra in (**e**). $1S_{HX,em}$ and $1S_{LX,em}$ are indicated in each panel. **g** PL spectra of $I_\parallel$ (upper panel) and $I_\perp$ (lower panel) at 0 degree (the grey box in (**e**)). Each spectrum is fitted with double Lorentzian curves to decouple $1S_{HX,em}$ (red line with shading) and $1S_{LX,em}$ (blue line with shading). **h** Rotation angle dependent degree of polarization (DOP) of PL (black open circle), and its components, $1S_{HX,em}$ (red open circle) and $1S_{LX,em}$ (blue open circle). Each DOP is fitted with a sine function (solid lines in corresponding colors). **i** Polar graph of $1S_{HX,em}$ (red) and $1S_{LX,em}$ (blue) divided by the sum of PL intensity. The $1S_{LX,em}$ intensity is multiplied by 2 for better visibility.

shell on CdSe core without a presence of stress-alleviating alloy layers enlarges the energy split between $|1S_{HX}\rangle$ and $|1S_{LX}\rangle$ ($\Delta E_{split} \geq 50$ meV) greater than both the thermal energy ($k_B T$) and decreases the spectral linewidths from each emission ($E_{FWHM} \leq 19$ meV for $1S_{HX,em}$ and $E_{FWHM} \leq 26$ meV for $1S_{LX,em}$) (Fig. 3c), enabling to spectrally resolve each PL emission at room temperature and examine their characteristics independently.

Specifically, with a single-dot spectroscopy set-up embedding a half-wave plate (HWP) and a Wollaston prism (WP) (Fig. 3d), we probe polarization characteristics of PL emission from $|1S_{HX}\rangle$ and $|1S_{LX}\rangle$ of an individual sg-QD (see Methods for a detailed experiment set-up). Figure 3e displays the traces of PL spectra from a CdSe ($r = 2.5$ nm)-ZnSe ($H = 5.0$ nm) sg-QD, which are gained while rotating the polarization direction of the incident light from 0 to 360 degrees with an

acquisition time of 100 ms for each degree. Here, two linearly polarized components with orthogonal polarization ($I_\parallel$ and $I_\perp$) are measured, and they are summed to construct the incoming luminance signal ($I_0$). A double-Lorentzian fitting is performed to evaluate the intensity changes in PL emission from $|1S_{HX}\rangle$ and $|1S_{LX}\rangle$ for each PL spectrum (Fig. 3f). The degree of polarization (DOP), defined by $(x_\parallel - x_\perp)/(x_\parallel + x_\perp)$, where x represents the intensity of each component, is calculated to evaluate the amplitude and periodicity of PL signal changes according to the polarization.

Strong suppression of blinking and spectral diffusion shown in $I_0$ (top panel of Fig. 3e) allows us to exclude the impact of QD charging and subsequent Auger recombination processes or Stark effect on the spectral changes in $I_\parallel$ and $I_\perp$. This in turn suggests that the intensity changes of PL emission from $|1S_{HX}\rangle$ and $|1S_{LX}\rangle$ in $I_\parallel$ and $I_\perp$ upon the rotating angle ($\theta$) relate to their inherent polarization characteristics. DOPs of $|1S_{HX}\rangle$ and $|1S_{LX}\rangle$ are sinusoidal functions having a period of 180 degrees and the phase difference is 90 degrees (Fig. 3g, h), indicating that both exciton states are polarized with orientations perpendicular to each other. The experimental results are in line with the theoretical prediction that the degeneracy lift of $|1S_{HX}\rangle$ and $|1S_{LX}\rangle$ arises from compressive stresses imposed along two distinct normal crystal directions[12,28], which explains that PL emission from $|1S_{HX}\rangle$ emanates from the 2D dipole on the *AB* plane, whereas PL emission from $|1S_{LX}\rangle$ comes from the 1D dipole along the *C* axis (Supplementary Fig. 15). Dominated by the emission from $|1S_{HX}\rangle$ over $|1S_{LX}\rangle$ following the Boltzmann relation, the overall single exciton PL emission from CdSe-ZnSe sg-QDs shows strong directivity toward the *C* axis.

The degeneracy lift of exciton states into two distinct ones having orthogonal polarization by means of anisotropic strain is not limited to the lowest exciton states, but also materialized in higher exciton states. Supplementary Fig. 16 presents the polarization characteristics of 1 P emission from a CdSe ($r = 2.5$ nm)-ZnSe ($H = 5.0$ nm) sg-QD. Due to the structural robustness by the thick uniform shell and suppressed Auger recombination, the sg-QD demonstrates stable PL emission under a high fluence cw laser excitation, permitting to perform spectra-resolved polarization characterization with increasing average exciton numbers, $\langle N \rangle_{cw}$, from 0.01 to 6.85 (the calculation method is detailed in Supplementary Note 6). Resembling 1S emission, 1 P emission shows a strong main peak from $1P_e \rightarrow 1P_{HH}$ transition (referred to as $1P_{HX}$) and a shoulder peak from $1P_e \rightarrow 1P_{LH}$ transition (referred to as $1P_{LX}$) with an energy split of 35 meV, whose intensities alter periodically upon the polarization angle. $1P_{HX,em}$ and $1P_{LX,em}$ bear a strong similarity to $1S_{HX,em}$ and $1S_{LX,em}$ in respects to the amplitude and orientation in DOPs, suggesting the degeneracy lift of the second lowest quantized states of heavy hole and light hole by the anisotropic strain.

## Sg-QDs with an expanded emission envelope and their photonic application

Stable luminescence with a narrow spectral linewidth makes sg-QDs ideally suited photon sources for use in information displays and lasers. These applications demand fine control over the emission envelope of sg-QDs to cover the entire visible region, which can be attained by means of size and composition control of the emitting core in core-shell QD systems on the validity of the mutually strained pseudomorphic heteroepitaxy. We demonstrate the emission wavelength tunability by steering the thermodynamic growth of ZnSe shell on CdSe or $Cd_{0.25}Zn_{0.75}Se$ cores having varying radii (Fig. 4). It is noted that $Cd_{0.25}Zn_{0.75}Se$ cores have the same crystal structure (wurtzite) as CdSe core and ZnSe shell (Supplementary Fig. 17), facilitating the coherent growth of wz ZnSe shell. Figure 4a demonstrates single exciton PL emission of individual sg-QDs, whose peak PL energies vary from 1.95 to 2.65 eV. All QDs show two distinct PL peaks due to the degeneracy lift of exciton states imposed by anisotropic stress. We

observe that strain-relieving defect is effectively suppressed for all sg-QDs, as evidenced with high PL QYs exceeding 90% (Fig. 4b). The asymmetric strain by the pseudomorphic growth of ZnSe shell lifts the degeneracy of QDs to decouple $1S_{HX,em}$ and $1S_{LX,em}$ (Fig. 4a and Supplementary Fig. 18), which are polarized with an orientation perpendicular to each other (Fig. 4d–f). Above experimental data coherently attest the effectiveness of the direct pseudomorphic growth of ZnSe shell to induce asymmetric strain in both CdSe or $Cd_{0.25}Zn_{0.75}Se$ cores that lifts degeneracy of exciton states.

It is noted that, despite the similar trends observed in the photophysical properties with compressive strain by means of increasing ZnSe shell thicknesses, the impact appears less dramatic for $Cd_{0.25}Zn_{0.75}Se$-ZnSe sg-QDs compared to CdSe-ZnSe sg-QDs. Specifically, the spectral linewidth of individual $Cd_{0.25}Zn_{0.75}Se$-ZnSe sg-QDs is measured to be 30 meV in average, which is 10 meV broader than CdSe-ZnSe sg-QDs (Fig. 4c), and the heavy hole-light hole energy split in $Cd_{0.25}Zn_{0.75}Se$-ZnSe sg-QDs is 14 meV smaller than CdSe-ZnSe sg-QDs (Supplementary Fig. 18). These are attributed to the reduction of the magnitude of compressive strain in the core due to the smaller lattice mismatch between $Cd_{0.25}Zn_{0.75}Se$ and ZnSe (i.e., −1.43 % (−1.86%) along [0002] ([1000]) axis in bulk[27]), which lessen the effect of the strain such as phonon hardening. In addition, the reduced energy barrier for charge carrier confinement within $Cd_{0.25}Zn_{0.75}Se$ core by the ZnSe shell leaves excitons to be susceptible to the changes in their environment, which could bring about the spectral linewidth broadening.

The figure of merits of our sg-QDs can be easily transplanted into a range of photonic applications. Figure 4g–j exemplifies electroluminescent (EL) devices implementing sg-QDs as the emissive materials. Here, EL devices are structured in a well-established inverted structure with hybrid charge transport layers employing ZnMgO nanoparticle electron transport layer and 4,4′-Bis(N-carbazolyl)−1,1′-biphenyl (CBP) hole transport layer. Benefitted from the narrow spectral linewidth, high efficiency and structural robustness of sg-QDs, resulting devices show narrow EL spectrum ($E_{FWHM} = 50$ meV), which is the record narrow value among all types of EL devices (Supplementary Fig. 8), a peak external quantum efficiency (EQE) of 21.6% close to the theoretical limit, and a peak brightness exceeding 420,000 cd/m².

Above experimental results signify that the asymmetric compressive strain is an additional knob that allows to maneuver the photophysical characteristics of QDs. Contrary to the quantum confinement effect[37] or surface/interface polarization[38,39] that alter the electronic energy levels of QDs, the asymmetric compressive strain reconfigures the exciton fine structure and exciton-phonon coupling and consequently awards accelerated radiative recombination rates, narrow spectral linewidths and directed emission to QDs. The outcomes promise to expand the efficiency limit of photonic applications implementing QDs. For example, the narrow spectral linewidth of sg-QDs enables to increase the capacity of deliverable information by the light sources, and the arranged dipole moment of sg-QDs boosts the out-coupling efficiency of photonic systems employing QD.

Apparent next step is finding a way to operate the asymmetric compressive strain in diversified QD material systems. Specifically, the narrow spectral linewidth in heavy-metal free elements, such as InP[10,39,40] or $Ag(In,Ga)S_2$ QDs[41,42], will bear immediate and significant impacts to display and lighting industries. Also, the processing to align transition dipoles of sg-QDs in solids is necessary to enhance the out-coupling efficiency. Anisotropy in shape of the shell[43] or facet dependent surface energy[23] is a potential solution to align the transition dipoles of sg-QDs. The development of heteroepitaxy chemistry for the coherent pseudomorphic growth of III-V or I-III-VI₂ semiconductors in a desired morphology will allow us to exploit the key characteristics of anisotropic compressive strain, stimulating their practical use in a range of photonic applications.

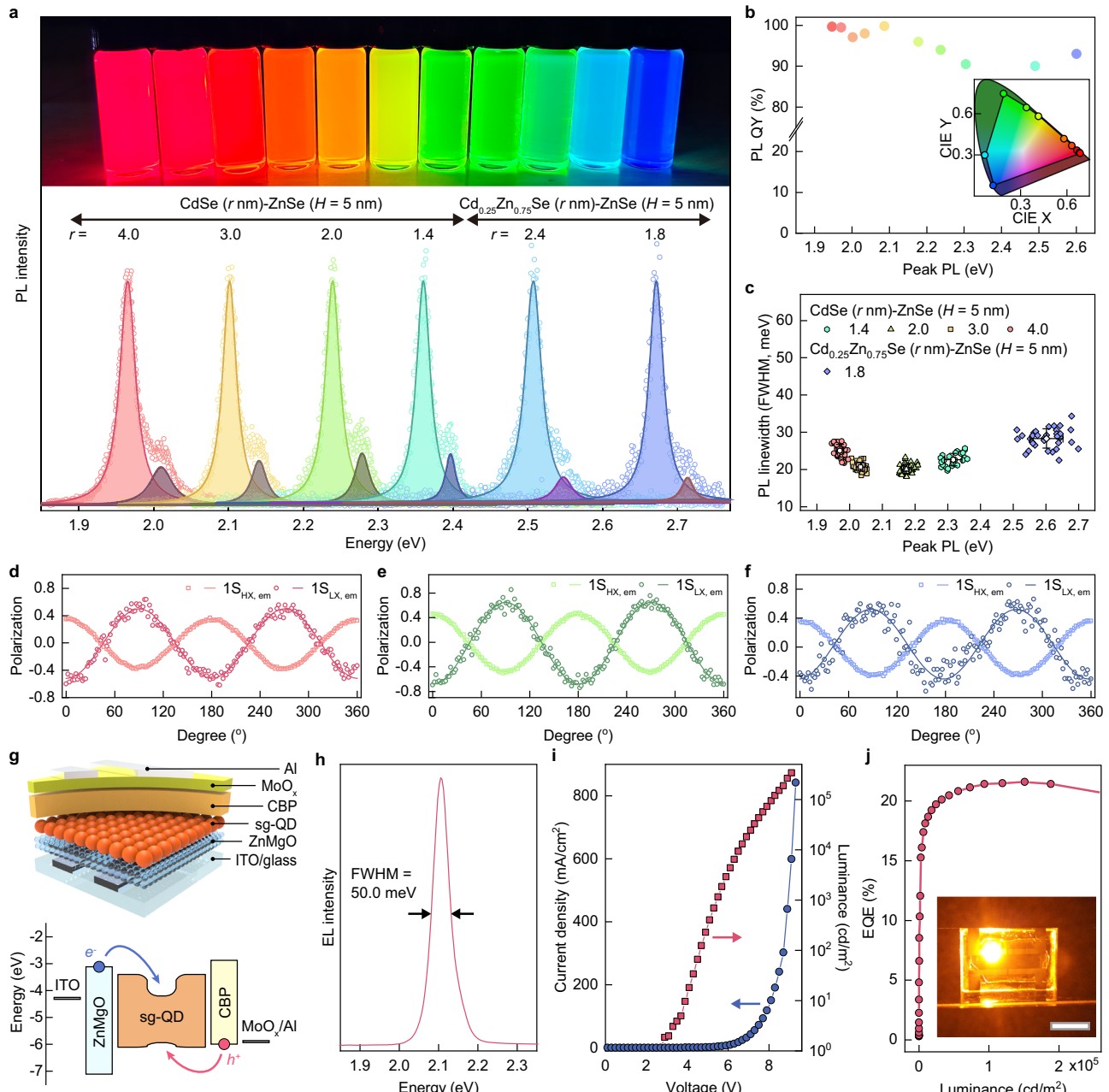

**Fig. 4 | sg-QDs with variable PL energies and their photonic application. a** Single exciton PL spectra of individual sg-CdSe ($r$, nm)-ZnSe ($H$ = 5.0 nm) and Cd$_{0.25}$Zn$_{0.75}$Se ($r$, nm)-ZnSe ($H$ = 5.0 nm) QDs with varying core radii. Each spectrum is fitted with double Lorentzian curves to decouple 1S$_{HX,em}$ (light shaded) and 1S$_{LX,em}$ (dark shaded). The inset is a photographic image of CdSe-ZnSe and Cd$_{0.25}$Zn$_{0.75}$Se-ZnSe sg-QD solutions with variable PL energies. **b** Absolute PL QYs of CdSe-ZnSe sg-QDs with varying core radii (1.4 nm ≤ $r$ ≤ 4.5 nm) and Cd$_{0.25}$Zn$_{0.75}$Se-ZnSe sg-QDs with varying core radii (1.8 nm ≤ $r$ ≤ 2.4 nm) at a fixed shell dimension ($H$ = 5.0 nm). The inset in (**b**) shows CIE color coordinates of the series of sg-QDs. **c** Statistics showing PL linewidths and peak PL energies of individual CdSe-ZnSe sg-QDs ($r$ = 4.0, 3.0, 2.0 and 1.4 nm, respectively) and Cd$_{0.25}$Zn$_{0.75}$Se-ZnSe sg-QDs ($r$ = 1.8 nm) at a fixed shell dimension ($H$ = 5.0 nm) (more than 25 dots per each sample). Symbols are averages and error bars are standard deviations. **d–f** Rotation angle dependent DOP of 1S$_{HX,em}$ and 1S$_{LX,em}$ for (**d**), red, (**e**), green and (**f**), blue emitting sg-QDs. **g** Schematic illustrations of QD-LED (top) and its energy band diagram (bottom). **h** EL spectrum, (**i**) current density-voltage-luminance characteristics, and (**j**) external quantum efficiencies (EQEs) *versus* luminance curve of the QD-LED (inset: a photograph of the working device, scale bar is 10 mm).

## Discussion

We have demonstrated strain-graded CdSe-ZnSe QDs in a centered core-shell geometry with compositionally abrupt interface. Specifically, we have devised heteroepitaxy chemistry to steer the coherent pseudomorphic growth of ZnSe epilayers on CdSe core, which renders strain-graded, defect-free core-shell heterostructured QDs with intensified asymmetric compressive strain and high structural homogeneity. Resulting sg-QDs feature degeneracy lift of exciton states, which involves the split of bright exciton states having polarization along the $C$ axis and normal to the $C$ axis and the decoupling of them from dark exciton states, resulting in the directed luminescence at an accelerated radiative recombination rate. Remarkably, sg-QDs show suppressed exciton-phonon coupling and reduced influence from the environmental charge fluctuation, which translate into the stable PL emission with near-unity PL QYs, record-narrow spectral linewidth (17.7 meV) and suppressed spectral diffusion (std = 0.27 meV) in an individual dot level (see the comparison with previously demonstrated compositionally-graded QDs in Supplementary Table 3). Plus,

benefitted from the structural fidelity, sg-QDs exhibit the record-setting spectral purity in both PL and EL, among colloidal core-shell nano-emitters in ensemble. By capitalizing on the asymmetric strain effect on various compositions of cores together with the quantum confinement effect, we have demonstrated sg-QDs with an expanded emission envelope spanning the entire visible light spectrum and the potential of our approach for photonic applications. The results imply that asymmetric strain is an effective means to award QDs exceptional spectral purity and efficiency and thus offer wider possibility for their practical use in a range of photonic applications.

## Methods

### Materials
Cadmium oxide (CdO, 99.99%), diphenylphosphine (DPP, 98%), oleylamine (Oam, 70%), 1-octadecene (ODE, 90%, technical grade), *n*-trioctylphosphine (TOP, technical grade, 90%), toluene (anhydrous 99.8%), hexane (anhydrous 95%), methanol (99.8%), ethanol ( > 99.8%) were purchased from Sigma-Aldrich. Zinc acetate (Zn(Ac)$_2$, 99.99%), oleic acid (OA, 99%), selenium (Se, 99.9%), sulfur (S, 99.9%) were purchased from Uniam. Benzoyl peroxide was purchased from Duksan. Molybdenum oxide (MoOx, 100 mesh powder, 99.995%) and the aluminum (5 mm Dia×5 mm Th pellets, Al, 99.999%) metal source were purchased from Taewon Scientific Co. (TASCO). (TASCO). 4,4′-Bis(N-carbazolyl)−1,1′-biphenyl (CBP, 99.9%) was purchased from OSM. All chemicals, unless otherwise stated, were used as received.

### Preparation of precursor solutions
All syntheses were performed under N$_2$ atmosphere using the Schlenk-line technique. 0.5 M cadmium oleate (Cd(OA)$_2$) and zinc oleate (Zn(OA)$_2$, Zn: OA = 1:3 (mol:mol)) stock solutions diluted in ODE were prepared for cation precursors and 2 M TOPS, TOPSe and 0.2 M DPPSe were prepared for anion precursors. For the preparation of Cd(OA)$_2$, 40 mmol of CdO and 80 mmol of OA were degassed at 110 °C for 1 h, and backfilled with N$_2$. The mixture was heated to 300 °C for 1 h, and then diluted with ODE to make 0.5 M concentration. For the preparation of Zn(OA)$_2$ with excess OA, 40 mmol of Zn(Ac)$_2$, 120 mmol of OA were degassed at 110 °C for 1 h, and backfilled with N$_2$. The mixture was heated to 280 °C, and then diluted with ODE to make 0.5 M concentration. For TOPSe preparation, 100 mmol of Se was mixed with 50 mL of TOP at 100 °C for 6 h in a glovebox. The same procedure was applied for the preparation of TOPS. DPPSe was prepared by mixing 6 mmol of Se with 3 mL of DPP at 200 °C for 5 min in a glovebox, and diluted to 0.2 M with toluene at RT.

### Synthesis of Se-terminated CdSe QDs with varying radii
To prepare 1.4 nm radius wz CdSe cores, 0.2 mmol of CdO, 0.4 mmol of OA and 3.5 ml of ODE were evacuated at 110 °C for 2 h. After back-filling with N$_2$, the reaction flask was heated up to at 300 °C. 0.2 ml of 2 M TOPSe mixed with 0.2 ml of Oam was rapidly injected into the reaction flask at the elevated temperature. After 5 min of reaction, the reaction flask was rapidly cooled to obtain 1.4 nm radius wz CdSe cores. For additional growth, desired amounts of Cd(OA)$_2$ and TOPSe (1:1.2 (mol:mol)) were injected dropwise with syringe pump (i.e., 0.15, 0.75, 1.9 mmol of Cd stock solution, and 0.18, 0.9, 2.3 mmol of Se stock solution for 2.0, 3.0 and 4.0 nm radius wz CdSe cores, respectively. injection rates for Cd and Se stock solution = 2 and 2.4 mmol/hr, respectively). The growth was suspended when the targeted size was reached. Resulting QDs were purified via precipitation/redispersion method (acetone/toluene) and dispersed in toluene for further experiments.

### Synthesis of CdSe-ZnSe QDs with varying shell thicknesses
For the pseudomorphic growth of ZnSe shell onto CdSe cores, we optimized the ZnSe growth condition as follows: zinc oleate with low reactivity (Zn:OA = 1:3 (mol:mol)), high growth temperature (340 °C), and layer-by-layer growth of ZnSe (0.5 nm for each step, 30 min per each step) (Supplementary Fig. 3). For each step of ZnSe growth, desired amounts of Zn and Se precursors were injected dropwise (in 30 sec) and the reaction temperature was maintained for 30 min to complete the reaction. The same procedures with different amounts of precursors were repeated for the continued shell growth. The reaction flask was cooled to room temperature when the targeted shell thickness was reached. In case it is needed to enhance the oxidation stability, 0.5 nm of ZnS exterior shell layer was additionally grown on CdSe-ZnSe QDs by adding Zn and S stock solutions following the same reaction scheme of ZnSe growth. Resulting QDs were purified via precipitation/redispersion method (acetone/toluene) and dispersed in toluene for further characterization.

### Synthesis of Cd0.25Zn0.75Se-ZnSe core-shell QDs
To prepare CdZnSe alloyed cores, 0.04 ml of Cd(OA)$_2$, 0.16 ml of Zn(OA)$_2$ and 6 ml of ODE were evacuated at 110 °C for 2 h. After back-filling the reaction flask with N$_2$, the temperature was heated up to 300 °C. 0.5 (0.75) ml of 0.2 M DPPSe was swiftly injected to produce 2.4 (1.8) nm radius wz CdZnSe cores. To adopt ZnSe shell onto CdZnSe cores, calculated amounts of Zn(OA)$_2$, and TOPSe were injected sequentially in every 30 min. The growth was suspended when the targeted shell thickness was achieved. To further improve the oxidation stability, an additional layer of 0.5 nm ZnS shell was grown on the Cd$_{0.25}$Zn$_{0.75}$Se-ZnSe QDs.

### Etching of CdSe-ZnSe core-shell QDs
Benzoyl peroxide was used as oxidative etchant. First, we prepared the etchant solution by dissolving 0.5–2.0 g of benzoyl peroxide in a mixed solution of 3 ml of toluene, 3 ml of hexane and 1 ml of methanol. 0.6 ml of QD dispersion in hexane (optical density of 2 at the first exciton peak) was swiftly injected into the etchant solution at room temperature to initiate the etching process. The extent of etching was controlled by varying the amount of benzoyl peroxide (0.5–2.0 g) and reaction time (1–4 h). The reaction was terminated by adding 2 ml of TOP and 20 ml of ethanol. Residual QDs were precipitated by centrifuging the entire solution for 5 min at 6000 rpm and collected by dispersing the precipitates with 5 ml of toluene.

### Device fabrications
For QD-LEDs fabrication (ITO/Zn$_{0.9}$Mg$_{0.1}$O/QDs/CBP/MoOx/Al), 20 mg/mL Zn$_{0.9}$Mg$_{0.1}$O nanoparticles were spun-cast on the ITO substrate at 4000 rpm for 30 s and annealed at 75 °C for 30 min in a glovebox. 14 mg/mL QDs were spun-cast at 4000 rpm for 30 s on ZnO/ITO and annealed at 75 °C for 30 min in the glovebox. CBP (60 nm), MoO$_x$ (10 nm), and Al (120 nm) were thermally deposited on the QDs/ZnO/ITO films under a pressure of -10$^{-6}$ Torr at a deposition rate of 1.0 − 1.5, 0.1 − 0.2, and 1.0 − 2.0 Å/s, respectively. QDs were used after purification without further ligand exchange process. The devices were encapsulated for subsequent characterization.

### Characterization
UV-Vis, PL and absolute PL QY measurements were conducted using the UV-1800 (Shimadzu), FluoroMax-4 (Horiba) and Quantaurus-QY plus (Hamamatsu photonics), respectively. HR-TEM images were obtained using the Talos F200i operating at 200 kV. Inverse FFT and geometric phase analysis was performed using Gatan Microscopy Suite Software. High-resolution X-ray diffraction (XRD) was conducted at 8 D beamline of the Pohang Accelerator Laboratory (PAL). The chemical composition of CdSe core was analyzed by ICP-AES (Perkin-Elmer, OPTIMA-4300DV). Single-dot measurements were conducted using the 405 nm (3.06 eV) excitation beam (PicoQuant, LDH-D-C-405 laser diode) in pulsed mode at a 1 MHz repetition rate for time-resolved PL dynamics measurements and cw mode for

polarization-resolved spectrum measurements. The laser beam was depolarized by passing through a depolarizer (EdmundOptics). A diluted solution was drop-casted onto the cover slip. The cover slip was encapsulated with UV-resin and another cover glass with an adhesive getter at the globe box. The depolarized laser beam was focused on the sample using the oil immersion objective (Olympus, UPLXAPO100XO, 1.45 NA), and the signals were collected through the same objective lens and directed to Hanbury Brown-Twiss setup comprising two single-photon avalanche diodes (Micro Photon Devices, PDM Series) connected with time-correlated single photon counting module (PicoQuant, HydraHarp 400) to measure the PL dynamics, blinking statistics, and the second order correlation function $g^{(2)}(t)$. The beam can also be guided to the EMCCD camera (Princeton Instruments, ProEM HS1024BX3) attached to the spectrometer (Princeton Instruments, IsoPlane SCT320) after passing through a half-wave plate mounted on a stepper motor and the Wollaston prism to resolve polarization. The depolarizer (Thorlabs, DPP25-A) was placed in front of the spectrometer to ensure that the detection efficiency is independent of the polarization component of the emission. The current density–voltage–luminance ($J$-$V$-$L$), EQE–luminance and electroluminescence spectra are obtained with a combined system of a Keithley 237 source-measurement unit, a Keithley-2000 multimeter and a Konica-Minolta spectroradiometer (CS-2000).

## Data availability

Source data are provided as a Source data file. Source data are provided with this paper.

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

## Acknowledgements

This work was supported by the National Research Foundation of Korea (NRF) funded by the Ministry of Science, ICT and Future Planning 2021M3H4A3A01062960 (W.K.B., and D.C.L.)), and RS-2024-00445116 (W.K.B.), the Ministry of Trade, Industry & Energy (MOTIE, Korea) (20010737, RS-2024-00440884 (W.K.B.) and 20019417 (E.H.)), Korean Institute for Advancement of Technology (KIAT) grant funded by the Korea Government (MOTIE) (00418086, HRD Program for Industrial Innovation and P0017305, Human Resource Development Program for Industrial Innovation (Global) (W.K.B.)) and Samsung Display (W.K.B.).

## Author contributions

W.K.B., D.J., and J.W.P. conceived the idea. J.W.P., S.M., D.S., S.I., J.L., K.H.K., J.A.C., X.B. conducted synthesis and structural characterization. D.J., G.M.K., D.C.L., Y.S.P., R.P. led spectroscopic analysis. E.H., J.-S.P. performed computational calculation. H.J.L., J.S.P. fabricated the device. All authors contributed to the manuscript preparation.

## Competing interests

The authors declare no competing interests.
