## [Peer Review File · Nature Communications]

Strain-graded quantum dots with spectrally pure, stable and polarized emissionReviewer #1 (Remarks to the Author):

The authors reported a synthesis strategy for strain-graded CdSe/ZnSe core-shell QDs by suppressing the formation of interfacial alloyed layers of mixed cations (CdZnSe). Compositionally abrupt growth of the ZnSe shell is achieved using Se-rich core QDs. Although there are differences in terms of slight tweaks of synthetic protocols or the ligands/metal-chalcogen precursor used, the optical properties are commonly expected. The demonstrated narrow emission linewidths, reduced blinking, and Auger recombination are impressive but not surprising. Overall, the advances are incremental, and I regret that I can't recommend its acceptance in the current form in Nature Communication.

See below my detailed comments:

- Asymmetrically strained core-shell quantum dots, including the structure and composition reported in this study (CdSe/ZnSe core-shell QDs), have been reported. It is not clear in the introduction what the key issues of the target QDs are that the authors are addressing.
- The author claims that using Se-rich wurtzite core QDs (Cd/Se = 1.20) enables abrupt shell growth. I wonder what would happen in the case of zinc-blend cores?
- Geometric phase analysis reported in 1e contradicts the compositional discontinuity at the interface. The author should provide HAADF-STEM analysis along with intensity maps of sublattices or EDS line scan data to quantitatively show the discontinuity at the interface.
- Upon growing the ZnSe shell (regardless of the thickness), inhomogeneous broadening in the PL appears, which contradicts previous reports on CdSe/ZnSe core-shell QDs (Chem. Mater. 2021, 33, 5, 1799–1810, Chem. Mater. 2023, 35, 17, 7049–7059). The author should explain this discrepancy.
- The authors claim that splitting of excitonic features in absorption spectra as well as PL is attributed to the compressive strain imposed in CdSe core by ZnSe shell rather than Cd-Zn inter-diffusion at the interface. However, nearly the same splitting in optical features is reported for CdSe/CdZnSe core-shell QDs, see Park et al Nature Materials volume 18, pages 249–255 (2019). The author should clarify this.
- On page 9, authors claim suppression of blinking, spectral diffusion and Auger recombination in strain-graded QDs. However, there is no comparison of the reported data versus previous best reports. The author should draw a quantitative comparison and add a summary in the discussion.
- In Figure 4, the emission wavelength is tuned by varying the size of core QDs from 1.95 to 2.35 eV, while beyond this range, the tunability is achieved by synthesizing Cd_{0.25}Zn_{0.75}Se/ZnSe alloyed core/shell QDs. Are these QDs strain-graded? If so, then doesn't this contradict the author's earlier narrative that strain-graded CdSe/ZnSe core-shell QDs were achieved by suppressing the formation of interfacial alloyed layers of mixed cation (CdZnSe)?
- In Figure 4a, why is the PL contribution from the light hole exciton state significantly lower in Cd_{0.25}Zn_{0.75}Se/ZnSe compared to the strain-graded CdSe/ZnSe QDs?
- The authors reported PLQY data in Figure 4-f. Is this an absolute PLQY or relative? The author should add more details in the experimental section. And why does PLQY drop in Cd_{0.25}Zn_{0.75}Se/ZnSe QDs?
- ZnSe is generally prone to oxidation. Can the author comment on the air stability of the reported core-shell QDs?
- Recent studies on ZnSeTe/ZnSe core-shell QDs show that the inhomogeneous broadening of PL is due to Te clustering in core QDs (<https://doi.org/10.21203/rs.3.rs-1183117/v1>, <https://doi.org/10.1002/adma.202303528>). A Se-rich interface would likely create a similar situation. The author should comment on this.
- The author reported interesting electroluminescence data on technically less appealing wavelengths. I wonder why the author chose this wavelength instead of pure red, green, or blue?

Reviewer #2 (Remarks to the Author):

This article of "Strain-graded quantum dots with spectrally pure, stable and polarized emission" analyzed how structural deformation influences optical properties of QDs and showed QDs with high efficiency, narrow emission width and various spectral wavelengths. Although previous literatures reported the change of bandgap and exciton fine structure and single dot spectroscopic analysis for asymmetrically strained core/shell QDs, the authors separated strain and alloying

effects by using compositionally abrupt but strain-graded core/shell QD interface. The compositional discontinuity at the QD interface was explained by reasonable methods and evidences and the compressive strain was quantified and controlled. Maximizing the strain of QDs improved their optical properties, and especially, their remarkable linewidths are narrower than perovskites and nanoplatelet. The record narrow FWHM was maintained as 50 meV in EL devices with high brightness, too. The authors proposed how to utilize a new parameter of compressive strain for controlling the photophysical properties of QDs. I would like to recommend this article as a meaningful result for the QD community.

However, I think the following parts need to be explained more clearly.

1. The paper of "Park, Y. S., Lim, J. & Klimov, V. I. Asymmetrically strained quantum dots with non-fluctuating single-dot emission spectra and subthermal room-temperature linewidths. *Nat. Mater.* 18, 249-255 (2019)" also described the photophysical properties of asymmetrically strained QDs. Please clarify how this article is different and new compared to the previous paper.
2. Asymmetric strain results in the splitting of exciton fine structure as heavy hole and light hole. I wonder if the significant difference of two emission states causes different stability. Do the linewidths, emission wavelengths, efficiency and energy difference between the split levels maintain during EL device operation?
3. The authors reconfirmed that the increased exciton energy is due to strain, not interface alloying using chemical etching. This part suddenly came up and was an unkind part to readers. I recommend to add more comments to the paper or the supplementary information.
4. In Supplementary Note 1, the caption needs to be corrected from (a) and (b) to (a) and (c).
5. I recommend the change of graphs in Supplementary Fig 13a and 13b from CdSe/ZnSe to CdZnSe core/shell QDs. Although the spectra of CdSe/ZnSe QDs appeared in the front part of the paper, CdZnSe core/shell QDs only appeared in Fig 4. Therefore, it would be better to show the graphs of CdZnSe core/shell QDs.

Reviewer #3 (Remarks to the Author):

The authors have developed defect-free core-shell Quantum Dots (QDs) that exhibit spectrally pure, stable, and polarized emission. This was achieved by introducing asymmetric compressive strain within the core region, which lifts the degeneracy of exciton states. Specifically, the authors synthesized CdSe-ZnSe QDs, reducing compositional and structural inhomogeneity by using a Se-rich CdSe core. This approach avoids unintentional alloying at the interface and reduces spectral broadening. As a result, the authors have attained near-unity photoluminescence quantum yields (PL QYs), narrow spectral linewidths, and suppressed spectral diffusion at the individual dot level, and high spectral purity in ensemble. Although some of finding has been published in the author's previous paper with Klimov, there are some interesting and new points here that are of interests to the readers. Especially, they demonstrated the feasibility of extending their approach to the entire visible spectrum. The reported approach can be helpful for a variety of photonic applications based on QDs, such as LED, laser, and single photon sources. Overall, the analysis and the experimental data support the author's hypothesis. I believe this work can be published after addressing the following concerns.

1. Asymmetrically stressed QDs with similar compositions but core/shells heterostructure have been reported (ref 27). The novelty of the authors' work lies in the synthesis of a compositionally abrupt interface, which reduces the spectral broadening originating from compositional and structural inhomogeneity. This pertains to the ensemble level. While the authors only present a comparison of the FWHM of individual QD in Fig. 1k, please provide ensemble-level comparisons. Additionally, the authors mention that compositional gradients reduce the magnitude of compressive stress, whereas abrupt compositional interfaces enhance the strain imposed in CdSe core. Please attach a comparison of these two QDs to substantiate this point.
2. It is noted that as the shell thickness increases, the effective compressive strain also increases, while the heavy hole-light hole energy split exhibits a decreasing trend after reaching its peak. The authors attribute this phenomenon to the growth of a bulk-like ZnSe shell. However, this explanation may not be readily understandable. In biaxially strained CdSe-CdS core-shell QDs reported by Fan et al., the strain is provided by asymmetric shell, when the shell grows thick, it is hard to avoid grow shell on the thin side of QD. However, the mechanism in ZnSe based strained

dots is different. Why the splitting also saturates after certain shell thickness? Further analysis or supplementation with additional DFT theoretical simulations would be beneficial to clarify this aspect.

3. The wurtzite/zinc blende polytypic structure has been considered as another possible reason to HH-LH splitting (Nat. Photon. 18, 186–191 (2024)). Could the authors please check their samples and identify whether this could another reason for the observed splitting? XRD and HRTEM will be helpful.

Response to Reviewers

Reviewer #1 (Publish after revision noted):

General comment: The authors reported a synthesis strategy for strain-graded CdSe/ZnSe core-shell QDs by suppressing the formation of interfacial alloyed layers of mixed cations (CdZnSe). Compositionally abrupt growth of the ZnSe shell is achieved using Se-rich core QDs. Although there are differences in terms of slight tweaks of synthetic protocols or the ligands/metal-chalcogen precursor used, the optical properties are commonly expected. The demonstrated narrow emission linewidths, reduced blinking, and Auger recombination are impressive but not surprising. Overall, the advances are incremental, and I regret that I can't recommend its acceptance in the current form in *Nature Communication*. See below my detailed comments.

Response: We sincerely appreciate Reviewer #1's constructive comments and suggestions on our present work. All authors thoroughly went over all comments and suggestions, prepared the responses, and revised the manuscript accordingly. Specifically, per reviewer's requests, we conduct comparative EDS elemental mapping analysis on CdSe-ZnSe QDs grown from Cd-rich *versus* Se-rich CdSe core (**Supplementary Fig. 2**) to support our claim on the composition profile across the core-shell interface. In addition, to strengthen novelty and advancement of our work in comparison with previous reports, we revise the introduction part of the manuscript to implement the novelty of our approach and our findings and summarize the comparison of key characteristics of our sg-QDs *versus* others in the revised version of Supplementary Information (**Supplementary Table. 3**). Below, we enclose the point-by-point responses to each comment raised by the Reviewer #1. We look forward that the revised version of manuscript sufficiently addresses all the issues and now suitable for publication in *Nature Communications*.

Comment #1-1: Asymmetrically strained core-shell quantum dots, including the structure and composition reported in this study (CdSe/ZnSe core-shell QDs), have been reported. It is not clear in the introduction what the key issues of the target QDs are that the authors are addressing.

Response: Our work contains novelty in the chemistry, materials, experimental findings and scientific discussion. Our research starts from new synthetic route, the coherent pseudomorphic growth of ZnSe shell on CdSe core within a symmetric core-shell geometry, which successfully navigates the challenges posed by the significant lattice mismatch between CdSe and ZnSe exceeding 7%. As the reviewer pointed out, the phenomenon (degeneracy lift in asymmetrically strained core-shell QDs) and the composition and structure (CdSe-ZnSe QDs) have been reported independently before, but **the degeneracy lift in CdSe-ZnSe QDs has not been demonstrated before!**

Here, we succeed to devise on-demand coherent pseudomorphic growth of ZnSe shell directly on CdSe core without creating misfit defects. The simplicity in QD structure and its controllability allow us to explore the relation between structural deformation and the photophysical properties of sg-QDs, which remains unsolved due to the structural complexity of QDs (e.g., CdSe-CdS with asymmetric shell geometry or CdSe-Cd_{1-x}Zn_xSe QDs having composition gradient) and the difficulty in controlling the structural parameters. In addition, the defect-free structure permits to avoid confusion coming from the structural complexity.

The comprehensive study together with computation enable us to rationalize 1) how the symmetric geometry brings about asymmetric strain, and 2) that the magnitude of strain, not the anisotropy itself, is responsible for the linewidth narrowing. Moreover, the present study for the first time reports the polarization characteristics of strained QDs. (Interestingly, the polarization characteristics are seen not only in 1S state but also in 1P state!) Most importantly, the present study devises effective means to impose full strain on QDs in the given compositional combination without creating misfit defects, allowing us to exploit the impact of the degeneracy lift in QDs, such as the record-narrow spectral linewidth at room temperature and an accelerated recombination rate (0.092 ns⁻¹) close to purely radiative one. As an ultimate achievement, we capitalize on the strain-graded structure together with the quantum confinement effect to expand emission envelope of sg-QDs spanning the entire visible region, providing new class of light sources with the increased capacity of deliverable information. These characteristics in comparison to CdSe-Cd_{1-x}Zn_xSe QDs with composition gradient (Nature Mater. 2019)) are summarized in **Table R1-1 (Supplementary Table 3)**.

Table R1-1 | Comparison of photophysical characteristics of strain-graded CdSe-ZnSe QDs and compositionally-graded CdSe-Cd_xZn_{1-x}Se QDs¹.

Key characteristics	Strain-graded QDs (this study)	Compositionally-graded QDs (ref. 1)
Structure	CdSe-ZnSe QDs with an abrupt interface	CdSe-Cd _x Zn _{1-x} Se QDs with composition gradient
PL QY	up to ~ 100 %	up to ~85 %
Ensemble PL linewidth	45 meV	68.6 meV
Single-dot PL linewidth	19.1 meV (in average)	24.1 meV (in average)
Radiative decay rate	0.092 ns ⁻¹	0.063 ns ⁻¹

Heavy hole-light hole splitting	42.8 meV	32.3 meV
Single-dot PL on-time fraction	95.80%	~ 95 %
Biexciton quantum yield*	7.3 % – 37.7 % (depending on the core radius)	~34 % (one example)
Spectral diffusion (standard deviation of PL peak energy)	0.65 meV	0.97 meV
PL Peak	460 nm - 640 nm	515 nm - 616 nm

*Biexciton quantum yields are calculated from following equation; $QY_{XX} = QY_X \times g^2(0)$.

Following the reviewer's suggestion, we have revised the introduction part of manuscript to clarify the aim and novelty of present work.

Change made in the manuscript

(page3, line 36-64)

Colloidal quantum dots (QDs) are free-standing nano-emitters that radiate photons of size-dependent adjustable wavelength with a narrow emission linewidth^{1,2}. Core-shell heterostructuring in these nanometer scale semiconductors allows one to tailor their photophysical characteristics³⁻¹⁰. Heteroepitaxy involves structural deformation of both core and shell materials in a way that the structural stress at the interface is efficiently alleviated. The structural deformation of core materials often appears to be enough to modify the electronic structure and photophysical characteristics¹¹⁻¹³. Specifically, asymmetric lattice strain imposed on QDs lifts the dark exciton states above the bright excitons and splits the bright excitons into two distinct states^{12,14,15}, accompanying the accelerated radiative decay, narrowed spectral linewidths and the reduced optical gain threshold which are crucial for QD applications in displays¹⁶⁻¹⁸, lasers¹⁹⁻²¹ and single-photon sources²².

The degeneracy lift in colloidal QDs was first reported from a spherical core-asymmetrically grown shell geometry¹⁹ and has been found even in a conventional spherical core-shell geometry^{16-18,21-24}. Despite progress, the relation between the structural factors and photophysical/electronic characteristics of results QDs remains unknown. The structural complexity of QD heterostructures, which includes an asymmetric geometry¹⁹ of shell or variations in composition^{16-18,20-23} or crystal structure²⁴ of shell along with misfit defects of near the interface, is the main culprit that impedes systematic investigation on the degeneracy lift in QDs. This calls for a materials platform that reaches degeneracy lift in core-shell heterostructure of a tailored geometry without misfit defect formation.

In the present study, we devise a coherent pseudomorphic growth technique for ZnSe shell over CdSe core to reach degeneracy lift in a conventional core-shell geometry. Resulting CdSe-ZnSe QDs bear compositionally abrupt but strain-graded core-shell interface, in which the compressive strain imposed in CdSe is greatly intensified while the formation of misfit defects is effectively suppressed. We perform comprehensive study across structural analysis, ensemble and single-dot spectroscopic analyses and computational calculations on the strain-graded QDs of variable core-shell dimensions, and relate the structural factors and their photophysical properties. Finally, we reveal the potential of coherent pseudomorphic growth of centered core-shell geometry to expand the emission energy envelope of strain-graded QDs, covering a wider spectral range and opening new avenues for applications in photonics.

Comment #1-2: The author claims that using Se-rich wurtzite core QDs ($\text{Cd/Se} = 1.20$) enables abrupt shell growth. I wonder what would happen in the case of zinc-blend cores?

Response: As Reviewer #1 suggested, we have grown zincblende ZnSe shell from the zincblende CdSe ($r = 2.5 \text{ nm}$) core having Se rich surface and compared photophysical characteristics with the case of zincblende CdSe ($r = 2.5 \text{ nm}$) core having Cd rich surface (**Fig. R1-1**). TEM with lattice stacking of ABCABC along the $\langle 111 \rangle$ direction confirms that resulting core-shell QDs are in zincblende structure (**Fig. R1-1a**). The difference between CdSe-ZnSe QDs having Se rich CdSe core *versus* Cd rich CdSe core is clearly seen in the changes of the absorption and PL energy upon ZnSe shell growth. Specifically, for Se rich case, the absorption and PL peak energies shift to higher energy immediately upon ZnSe growth (**Fig. R1-1b,d,f**). By contrast, Cd rich CdSe QDs show absorption and PL energy shift toward lower energy for 1 nm ZnSe shell growth and move to higher energy afterward (**Fig. R1-1c,e,f**). The increase in absorption and PL energy is a result of the compressive strain imposed on CdSe core by ZnSe shell. The PL energy shift to lower energy seen for the case of Cd rich CdSe core with 1 nm-thick ZnSe shell is attributed to the formation of CdZnSe alloy layer at the interfacial layer, which provides smaller potential wall than ZnSe, leading to the reduction of the confinement energy for charge carriers.

Fig. R1-1 | (a) HR-TEM image of zincblende CdSe ($r = 2.5 \text{ nm}$)-ZnSe ($H = 3.0 \text{ nm}$). Each line indicates the stacking along $[111]$, $[\bar{1}\bar{1}\bar{1}]$ with ABCABC, evidencing the zincblende structure along the core-shell geometry. (b,c) Absorption spectrum, (d,e) PL spectrum of (b,d) Se rich CdSe *versus* (c,e) Cd rich CdSe according to the ZnSe shell thickness ($H = 0, 1, 2, 3 \text{ nm}$). (f) $\Delta E_{\text{PL peak}}$ for the Cd rich CdSe *versus* Se rich CdSe according to the ZnSe shell thickness.

Comment #1-3: Geometric phase analysis reported in 1e contradicts the compositional discontinuity at the interface. The author should provide HAADF-STEM analysis along with intensity maps of sublattices or EDS line scan data to quantitatively show the discontinuity at the interface.

Response: Reviewer #1 suggested EDS line scan to quantitatively show the discontinuity of our core-shell QDs at the interface. After multiple runs of experiments, we have reached to the conclusion that EDS line scan, which shows projected 2D images of 3D spherical particles, is not effective to clearly discern the interface in an atomic scale². Instead, by comparing the radial composition distribution across the interface, we could tell the compositional discreteness of cations (Cd) at the interface in CdSe ($r = 2.5$ nm)-ZnSe ($h = 5.0$ nm) core-shell sg-QDs grown from CdSe core having Se rich surface (Se-rich CdSe-ZnSe, red) (**Fig. R1-2a,c**) in comparison to the ones grown from CdSe core having Cd rich surface (Cd-rich CdSe-ZnSe, black) (**Fig. R1-2b,c**). Considering that sg-QDs grown from CdSe core having Cd rich surface have a monolayer of CdZnSe unintentional alloy at the interface, which does not exist in the Se rich counterpart, the sharp contrast seen in Cd atom distribution at the interfaces strongly suggests that CdSe-ZnSe sg-QDs grown from Se rich CdSe has the compositional discontinuity of cations at the interface.

In response to the reviewer's feedback, we have added the experimental data in the revised version of Supplementary Information (**Supplementary Fig. 2**).

Fig. R1-2 | EDS elemental mapping of CdSe ($r = 2.5$ nm)-ZnSe ($h = 5.0$ nm) core-shell sg-QDs grown from (a) CdSe core having Se rich surface *versus* (b) CdSe core having Cd rich surface. (c) Averaged radial distribution of Cd atom for 5 individual QDs extracted from (a) (red) and (b) (blue). Black lines represent linear fits to the curvature. Their slopes are noted in figure.

Comment #1-4: Upon growing the ZnSe shell (regardless of the thickness), inhomogeneous broadening in the PL appears, which contradicts previous reports on CdSe/ZnSe core-shell QDs (Chem. Mater. 2021, 33, 5, 1799–1810, Chem. Mater. 2023, 35, 17, 7049–7059). The author should explain this discrepancy.

Response: Reviewer points out that upon ZnSe shell growth the inhomogeneous broadening in PL appears (exacerbates) in our samples. Also, the reviewer concerns that this phenomenon contradicts to previous reports on CdSe-ZnSe QDs. After careful review of our data and others in the references, we have reached to conclusion that, contrary to reviewer’s concern, inhomogeneous PL broadening diminishes (at least does not exacerbate) upon ZnSe shell growth in our samples.

We first checked 2 references, in which Reviewer #1 denoted that core-shell QDs free from inhomogeneous broadening were reported. In the first paper (Chem. Mater. 2021, 33, 1799-1810), PL linewidths (FWHMs) are 10.2 nm in ensemble, and 5.2 nm in single dot level for blue QDs and 16.3 nm in ensemble and 9.7 nm in single dot level for green QDs³. The discrepancy between ensemble and single dot level FWHMs represents inhomogeneous broadening in their samples. The second paper (Chem. Mater. 2023, 35, 7049-7059) only reports FWHM in ensemble (no single dot PL measurement), so the discussion on the inhomogeneous broadening is not made⁴.

Back to our work, we presented ensemble PL emission linewidth of CdSe-ZnSe sg-QDs with varying core radii ($2.0 \text{ nm} \leq r \leq 4.0 \text{ nm}$) and ZnSe shell thicknesses ($0 \text{ nm} \leq H \leq 5.0 \text{ nm}$) (**Fig. 2f**). In this figure, the PL linewidth continues to decrease upon ZnSe shell growth regardless of the core dimension. Statistics gained from single dot PL measurement show that the dot-to-dot variation in peak PL energy decreases for CdSe-ZnSe QDs having a thicker ZnSe shell (**Fig. R1-3**), indicating that the inhomogeneous PL broadening diminishes (at least does not becomes greater) upon ZnSe shell growth.

Fig. R1-3 | Individual dot-to-dot peak PL energy deviation for CdSe ($r = 2.5 \text{ nm}$) according to the ZnSe shell thickness.

Comment #1-5: The authors claim that splitting of excitonic features in absorption spectra as well as PL is attributed to the compressive strain imposed in CdSe core by ZnSe shell rather than Cd-Zn inter-diffusion at the interface. However, nearly the same splitting in optical features is reported for CdSe/CdZnSe core-shell QDs, see Park et al Nature Materials volume 18, pages 249–255 (2019). The author should clarify this.

Response: In the original manuscript, we attributed the reason for the separation of excitonic states into two distinct peaks in absorption and PL spectra of our sg-QDs to the splitting of heavy hole-light hole excitonic states by the compressive strain by citing the reference (Nat. Mater. 2019).

(page 6, line 112-114) As reported in previous studies of asymmetric QDs⁸⁻¹⁴, our sg-QDs features the degeneracy lift of exciton states, as signified with the energy split of the first exciton states in absorption and PL spectra and accelerated radiative recombination rates (Fig. 1f-i.)

The asymmetric strain imposed on QDs lifts the dark exciton states above the bright excitons and splits the bright excitons into two distinct states, accompanying the acceleration of radiative decay and spectral linewidth narrowing of QDs. The outcomes are of particular interest, as they are key characteristics to displays, lasers, and single-photon sources using QDs. The degeneracy lift in colloidal QDs was first reported in the spherical core-asymmetrically grown shell geometry by Fan & Sargent et al. (Nature 2017). After the first demonstration, QD community has moved forward to control the anisotropic strain given to QDs so as to exploit the characteristics of degeneracy lifted QDs. Nature Materials paper (2019) by Park & Klimov et al., is the milestone that reports the degeneracy-lift in a conventional spherical core-shell geometry. However, the relationship between the structural factors (magnitude of strain, and structural asymmetry) and photophysical/electronic characteristics of results QDs (the extend of degeneracy lift, linewidth narrowing, acceleration of radiative decay, Auger rates, etc) remains unknown due to the structural complexity and difficulty in controlling the structure of their samples. Therefore, the full control over the degeneracy-lift and related photophysical characteristics has not been reached.

In the present study, we succeeded to impose strain in a simple core-shell heterostructure in a symmetric geometry. On-demand growth of coherent pseudomorphic ZnSe shell on CdSe (or CdZnSe) core of varying dimensions allows us to relate the structural parameters that dictates the strain imposed on core QDs and photophysical characteristics of resulting QDs. In addition, the defect-free structure permits to avoid confusion coming from the structural complexity. As a result, together with validation of previously known relationship between the anisotropic strain and degeneracy lift, the systematic study enables us to rationalize 1) how the symmetric geometry brings about asymmetric strain, and 2) that the magnitude of strain, not the anisotropy itself, is responsible for the linewidth narrowing. Moreover, the present study for the first time reports the polarization characteristics of strained QDs. Most importantly, the present study devises effective means to impose full strain on QDs in given compositional combination without creating misfit defects, allowing us to exploit the impact of the degeneracy lift in QDs, such as the record-narrow spectral linewidth at room temperature and an accelerated recombination rate (0.092 ns^{-1}) close to purely radiative one. As an ultimate achievement, we capitalize on the strain-graded structure together with the quantum confinement effect to expand emission envelope of sg-QDs spanning the entire visible region, providing new class of light sources with the increased capacity of deliverable information.

The strong degeneracy lift in our sg QDs ($\Delta E_{split} = 42.8 \text{ meV}$) far exceeds that of previously reported one ($\Delta E_{split} = 32.3 \text{ meV}$)¹. In addition, the reduced compositional inhomogeneity in our samples aids to suppresses the inhomogeneous broadening in ensemble. These are orchestrated to allow us to clearly observe the splitting of excitonic states even in ensemble level absorption and PL spectra. We note that the distinction of heavy hole-light hole excitonic states in absorption spectra has not been seen before.

Comment #1-6: On page 9, authors claim suppression of blinking, spectral diffusion and Auger recombination in strain-graded QDs. However, there is no comparison of the reported data versus previous best reports. The author should draw a quantitative comparison and add a summary in the discussion.

Response: In response to the reviewer's comment, we have summarized the comparison of key photophysical characteristics of our sg-QDs with others reported in the previous study (**Table R1-1**). We have added this content to the discussion of manuscript and Supplementary Information (**Supplementary Table 3**). This comparison quantitatively highlights competitiveness of our strain-graded QDs over others.

Table R2-1 | Comparison of photophysical characteristics of strain-graded CdSe-ZnSe QDs and compositionally-graded CdSe-Cd_xZn_{1-x}Se QDs¹.

Key characteristics	Strain-graded QDs (this study)	Compositionally-graded QDs (ref. 1)
Structure	CdSe-ZnSe QDs with an abrupt interface	CdSe-Cd _x Zn _{1-x} Se QDs with composition gradient
PL QY	up to ~ 100 %	up to ~85 %
Ensemble PL linewidth	45 meV	68.6 meV
Single-dot PL linewidth	19.1 meV (in average)	24.1 meV (in average)
Radiative decay rate	0.092 ns ⁻¹	0.063 ns ⁻¹
Heavy hole-light hole splitting	42.8 meV	32.3 meV
Single-dot PL on-time fraction	95.80%	~ 95 %
Biexciton quantum yield*	7.3 % – 37.7 % (depending on the core radius)	~34 % (one example)
Spectral diffusion (standard deviation of PL peak energy)	0.65 meV	0.97 meV
PL Peak	460 nm - 640 nm	515 nm - 616 nm

*Biexciton quantum yields are calculated from following equation; $QY_{XX} = QY_X \times g^2(0)$.

Comment #1-7: In Figure 4, the emission wavelength is tuned by varying the size of core QDs from 1.95 to 2.35 eV, while beyond this range, the tunability is achieved by synthesizing Cd_{0.25}Zn_{0.75}Se/ZnSe alloyed core/shell QDs. Are these QDs strain-graded? If so, then doesn't this contradict the author's earlier narrative that strain-graded CdSe/ZnSe core-shell QDs were achieved by suppressing the formation of interfacial alloyed layers of mixed cation (CdZnSe)?

Response: Reviewer#1 concerns that the case studies of CdSe-ZnSe and CdZnSe-ZnSe QDs could contradict. We respectfully disagree with reviewer's point.

Our strategy to impose strong strain on the core is growing an abrupt shell without intentional or unintentional alloyed interfaces. For the combination of CdSe core with ZnSe shell, we terminate the surface of CdSe core with Se (Se-rich surface) and prompt pseudomorphic growth of ZnSe shell, so the formation of unintentional CdZnSe alloy layer is avoided. In this chosen combination, we could tune the peak PL energy of resulting core-shell QDs from 1.95 to 2.35 eV by varying the CdSe core size. To further push the energy envelope to a higher energy regime, we employ CdZnSe alloy core and take the same strategy, *i.e.*, the synthesis of Se-terminated CdZnSe core and passivation it with ZnSe shell. CdZnSe-ZnSe QDs also show degeneracy lift in excitonic states, as evinced with two distinct heavy hole and light hole excitonic states having polarized emission, indicating that resulting QDs are indeed asymmetrically strained.

Although the impact of strain imposed in CdZnSe core appears to be less effective than the case with CdSe core due to the reduced lattice mismatch between ZnSe and CdZnSe, these two case studies are still consistent in respects to the structural design, *i.e.*, starting from Se-terminated CdSe or CdZnSe core and passivation of ZnSe shell to avoid unexpected formation of cation mixed interface layers, and the outcomes, degeneracy lift of excitonic states. In addition, the case study with CdZnSe-ZnSe QDs offers the applicability of sg-QDs in a wide range of applications that demand pure, stable emission of higher energy photons. Given the reasons above, we adhere to include the experimental data of CdZnSe-ZnSe QDs and discussion in the manuscript and Supplementary Information.

Comment #1-8: In Figure 4a, why is the PL contribution from the light hole exciton state significantly lower in Cd_{0.25}Zn_{0.75}Se/ZnSe compared to the strain-graded CdSe/ZnSe QDs?

Response: As the reviewer point out, the “unintentionally chosen” PL spectrum of Cd_{0.25}Zn_{0.75}Se-ZnSe QDs show less strong PL from light hole exciton state than the CdSe-ZnSe QD samples. In fact, in a single-dot PL spectrum, the PL intensity ratio between heavy-hole and light-hole exciton state varies upon the orientation of QDs laying on a substrate (**Supplementary Fig. 11**), and, thus, statistics of single-dot level characteristics are required to comprehend this fairly. We have added more examples of Cd_{0.25}Zn_{0.75}Se-ZnSe QDs in the **Supplementary Fig. 14**, which includes single-dot PL spectrum of a Cd_{0.25}Zn_{0.75}Se-ZnSe QD with stronger PL from light hole exciton state.

Supplementary Fig. 1 | Schematic representations of the transition dipole moments in the sg-QD and the polarization of their emission with projection onto the observation plane. (a) $1S_{HX,em}$ emission from 2D dipole on the AB plane of wurtzite crystal and (b) $1S_{LX,em}$ emission from 1D dipole along the C axis of wurtzite crystal. (c) The exciton fine structure of wurtzite CdSe with prolate spheroid shape⁵. For simplicity, only bright states are denoted. $|1S_{HX}\rangle$ is composed of $\pm 1^L$ which exhibit 2D dipole on the AB plane, while $|1S_{LX}\rangle$ is composed with 0^U , which demonstrates 1D dipole along the C axis, and $\pm 1^U$, which exhibit 2D dipole on the AB plane. Because the oscillator strength of 0^U is higher than that of $\pm 1^U$, the emission from the $|1S_{LX}\rangle$ is primarily dominated by 0^U component, supporting the dipole direction along the C axis for $1S_{LX,em}$.

Change made in Supplementary Information

(Supplementary Fig. 14)

Supplementary Fig. 14 | (a) PL emission spectrum (grey dot) and its component, $1S_{HX,em}$ (red shading) and $1S_{LX,em}$ (blue shading), of a single $\text{Cd}_{0.25}\text{Zn}_{0.75}\text{Se}$ ($r = 1.8 \text{ nm}$)-ZnSe ($H = 5.0 \text{ nm}$) sg-QD. Emission linewidth for PL spectrum and the energy splitting are noted. (b) Histogram showing the energy split ($1S_{HX,em} - 1S_{LX,em}$) gained from 38 individual $\text{Cd}_{0.25}\text{Zn}_{0.75}\text{Se}$ ($r = 1.8 \text{ nm}$)-ZnSe ($H = 5.0 \text{ nm}$) sg-QDs. The mean split energy is ca. 28.0 meV. (c) $1S_{HX,em} - 1S_{LX,em}$ energy splits for the CdSe ($1.4 \text{ nm} \leq r \leq 4.5 \text{ nm}$)-ZnSe ($H = 5.0 \text{ nm}$) sg-QDs and $\text{Cd}_{0.25}\text{Zn}_{0.75}\text{Se}$ ($1.8 \text{ nm} \leq r \leq 2.4 \text{ nm}$)-ZnSe ($H = 5.0 \text{ nm}$) sg-QDs.

Comment #1-9: The authors reported PLQY data in Figure 4-f. Is this an absolute PLQY or relative? The author should add more details in the experimental section. And why does PLQY drop in Cd_{0.25}Zn_{0.75}Se/ZnSe QDs?

Response: PL QYs in Figure 4b are absolute values measured with Quantaaurus-QY plus (Hamamatsu photonics). We clarify this in the revised version of manuscript.

We speculate that the diminished energy barrier for charge carrier confinement into Cd_{0.25}Zn_{0.75}Se core by ZnSe shell in comparison to CdSe-ZnSe cases is responsible for the reduction of PL QYs. We would like to point out that, despite PL QYs of Cd_{0.25}Zn_{0.75}Se-ZnSe QDs seem inferior compared to these of CdSe-ZnSe QDs, they are still around 90 % close to the unity.

Change made in the manuscript

(page30, caption of Fig. 4b)

Fig. 4 sg-QDs with variable PL energies and their photonic application. a. Single exciton PL spectra of individual sg-CdSe (*r*, nm)-ZnSe (*H* = 5.0 nm) and Cd_{0.25}Zn_{0.75}Se (*r*, nm)-ZnSe (*H* = 5.0 nm) QDs with varying core radii. Each spectrum is fitted with double Lorentzian curves to decouple 1S_{HX,em} (light shaded) and 1S_{LX,em} (dark shaded). The inset is a photographic image of CdSe-ZnSe and Cd_{0.25}Zn_{0.75}Se-

ZnSe sg-QD solutions with variable PL energies. **b**, Absolute PL QYs of CdSe-ZnSe sg-QDs with varying core radii ($1.4 \text{ nm} \leq r \leq 4.5 \text{ nm}$) and $\text{Cd}_{0.25}\text{Zn}_{0.75}\text{Se}$ -ZnSe sg-QDs with varying core radii ($1.8 \text{ nm} \leq r \leq 2.4 \text{ nm}$) at a fixed shell dimension ($H = 5.0 \text{ nm}$). The inset in **b** shows CIE color coordinates of the series of sg-QDs. **c**, Statistics showing PL linewidths and peak PL energies of individual CdSe-ZnSe sg-QDs ($r = 4.0, 3.0, 2.0$ and 1.4 nm , respectively) and $\text{Cd}_{0.25}\text{Zn}_{0.75}\text{Se}$ -ZnSe sg-QDs ($r = 1.8 \text{ nm}$) at a fixed shell dimension ($H = 5.0 \text{ nm}$) (more than 25 dots per each sample). **d-f**, Rotation angle dependent DOP of $1S_{HX,em}$ and $1S_{LX,em}$ for **d**, red, **e**, green and **f**, blue emitting sg-QDs. **g**. Schematic illustrations of QD-LED (top) and its energy band diagram (bottom). **h**. EL spectrum, **i**. current density-voltage-luminance characteristics, and **j**. external quantum efficiencies (EQEs) versus luminance curve of the QD-LED (inset: a photograph of the working device).

(Page 17, line 358)

Characterization. UV-Vis, PL and absolute PL QY measurements were conducted using the UV-1800 (Shimadzu), FluoroMax-4 (Horiba) and Quantaaurus-QY plus (Hamamatsu photonics), respectively.

Comment #1-10: ZnSe is generally prone to oxidation. Can the author comment on the air stability of the reported core-shell QDs?

Response: As the reviewer points out, ZnSe is easy to oxidize and irreversibly degraded under air exposure. This becomes important for QDs' application to devices that include deposition processing of QD films in air. For application purposes, to keep the photophysical stability against oxidation, we passivated the surface of CdSe-ZnSe sg-QDs with a thin ZnS exterior shell layer (approximately 0.5 nm), as stated in **Methods** section of the original version of manuscript. We note that the thin ZnS exterior layer does not affect the photophysical characteristics of QDs.

Comment #1-11: Recent studies on ZnSeTe/ZnSe core-shell QDs show that the inhomogeneous broadening of PL is due to Te clustering in core QDs (<https://doi.org/10.21203/rs.3.rs-1183117/v1>, <https://doi.org/10.1002/adma.202303528>). A Se-rich interface would likely create a similar situation. The author should comment on this.

Response: ZnSeTe is a highly mismatch alloy, whereby significant difference in the electronegativity between Se and Te is present⁶⁻⁸. In a dilute Te concentration in ZnSe, Te clustering creates localized hole trap state below the valence band edge energy level with an energy gap of approximately 150-200 meV depending on the number of Te clusters and their dielectric environment. The morphological inhomogeneity among ZnSeTe QDs causes large energy disparity among individual ZnSeTe QDs, leading to the significant PL spectra broadening in ensemble.

By contrast, CdSe-ZnSe sg-QDs consist of the same anion (Se) throughout the core and shell, and the electronegativity between cations (Cd versus Zn) is not large enough to bring about such effects. Only the potential profile across the CdSe core and ZnSe shell determines the PL energy as other typical QD systems.

Comment #1-12: The author reported interesting electroluminescence data on technically less appealing wavelengths. I wonder why the author chose this wavelength instead of pure red, green, or blue?

Response: We choose CdSe ($r = 2.5$ nm)-ZnSe sg-QDs as the representative, which show distinct excitonic states from heavy-hole and light-hole in absorption and PL emission spectra in ensemble, and try to be consistent in making up the figures.

Reviewer #2 (Publish after revision noted):

General Comment: This article of “Strain-graded quantum dots with spectrally pure, stable and polarized emission” analyzed how structural deformation influences optical properties of QDs and showed QDs with high efficiency, narrow emission width and various spectral wavelengths. Although previous literatures reported the change of bandgap and exciton fine structure and single dot spectroscopic analysis for asymmetrically strained core/shell QDs, the authors separated strain and alloying effects by using compositionally abrupt but strain-graded core/shell QD interface. The compositional discontinuity at the QD interface was explained by reasonable methods and evidences and the compressive strain was quantified and controlled. Maximizing the strain of QDs improved their optical properties, and especially, their remarkable linewidths are narrower than perovskites and nanoplatelet. The record narrow FWHM was maintained as 50 meV in EL devices with high brightness, too. The authors proposed how to utilize a new parameter of compressive strain for controlling the photophysical properties of QDs. I would like to recommend this article as a meaningful result for the QD community. However, I think the following parts need to be explained more clearly.

Response: We are pleased with Reviewer #2’s positive evaluation (Publish after revision noted), constructive comments and suggestions on our work. All authors meticulously reflected all comments and suggestions, prepared the response letter, and revised the manuscript accordingly. Specifically, per Review #2’s comments, we clarify our study and add the extent to which our study differs from the previous research, provide complementary data set to show the EL device properties during the operation and detail the etching experiment with revised manuscript and supplementary information. Below are the point-by-point responses to the reviewer’s comments. We anticipate that the revised version of manuscript is now suitable for publication in *Nature Communications*.

Comment #2-1: The paper of “Park, Y. S., Lim, J. & Klimov, V. I. Asymmetrically strained quantum dots with non-fluctuating single-dot emission spectra and subthermal room-temperature linewidths. *Nat. Mater.* 18, 249-255 (2019)” also described the photophysical properties of asymmetrically strained QDs. Please clarify how this article is different and new compared to the previous paper.

Response: The asymmetric strain imposed on QDs lifts the dark exciton states above the bright excitons and splits the bright excitons into two distinct states, accompanying the acceleration of radiative decay and spectral linewidth narrowing of QDs. The outcomes are of particular interest, as they are key characteristics to displays, lasers, and single-photon sources using QDs. The degeneracy lift in colloidal QDs was first reported in the spherical core-asymmetrically grown shell geometry by Fan & Sargent et al. (*Nature* 2017). After the first demonstration, QD community has moved forward to control the anisotropic strain given to QDs so as to exploit the characteristics of degeneracy lifted QDs. *Nature Materials* paper (2019) by Park & Klimov et al., is one of the milestones that reports the degeneracy-lift in a conventional spherical core-shell geometry. However, the relationship between the structural factors (magnitude of strain, and structural asymmetry) and photophysical/electronic characteristics of results QDs (the extend of degeneracy lift, linewidth narrowing, acceleration of radiative decay, Auger rates, etc) remains unknown due to the structural complexity and difficulty in controlling the structure of their samples. Accordingly, the full control over the degeneracy-lift and related photophysical characteristics has not been reached.

In our research, we have successfully developed a method for the coherent pseudomorphic growth of a ZnSe shell over a CdSe core, executed within a symmetric core-shell architecture that spans various dimensions, all while bypassing the formation of misfit defects. This breakthrough in QD design and its precise manipulation unveils new possibilities to investigate how structural modifications influence the electronic and photophysical behavior of sg-QDs. Such explorations have previously been hindered by the intricate nature of QDs, seen in CdSe-Cd_{1-x}Zn_xSe QDs marked by compositional gradients, compounded by the challenges in managing structural parameters. Moreover, our approach's ability to maintain a defect-free structure helps clear the obscurities typically introduced by the inherent complexity of QDs.

Our investigation extends beyond the methodology to offer new insights into the photophysical phenomena resulting from the strain. We report, for the first time, the polarization characteristics of strained QDs observable in both the 1S and 1P states, a phenomenon not previously documented. This discovery stems from our ability to impose a full strain on QDs in a controlled manner, highlighting the relationship between structural parameters and its photophysical outcomes. The systematic study sheds light on how symmetric geometry induces asymmetric strain effects, leading to significant findings such as linewidth narrowing governed by the magnitude of strain rather than its anisotropy. These results underscore the uniqueness of our approach and its contribution to advancing the understanding of quantum dot science, particularly in relation to their electronic structure and optical properties.

Furthermore, the marked degeneracy lift ($\Delta E_{\text{split}} = 42.8 \text{ meV}$) observed in our sg-QDs not only surpasses previously reported values but also demonstrates our success in reducing compositional inhomogeneity, thereby mitigating the inhomogeneous broadening in ensemble measurements. This achievement allows for the distinct observation of the splitting of excitonic states in absorption and PL spectra, a clear distinction from prior works. Our comprehensive analysis, which includes a comparison of key photophysical characteristics of CdSe-ZnSe sg-QDs with those of QDs with composition gradients, presents a significant leap forward in the field (see detailed comparison of key characteristics between our CdSe-ZnSe sg-QDs *versus* compositionally-graded QDs). By addressing and overcoming the challenges previously faced by researchers in this domain, our study paves the way for new applications and enhanced functionalities of these special types of QDs, firmly establishing its novelty and impact in comparison to existing literature.

Table R2-1 | Comparison of photophysical characteristics of strain-graded CdSe-ZnSe QDs and compositionally-graded CdSe-Cd_xZn_{1-x}Se QDs¹

Key characteristics	Strain-graded QDs (this study)	Compositionally-graded QDs (ref. 1)
Structure	CdSe-ZnSe QDs with an abrupt interface	CdSe-Cd _x Zn _{1-x} Se QDs with composition gradient
PL QY	up to ~ 100 %	up to ~85 %
Ensemble PL linewidth	45 meV	68.6 meV
Single-dot PL linewidth	19.1 meV (in average)	24.1 meV (in average)
Radiative decay rate	0.092 ns ⁻¹	0.063 ns ⁻¹
Heavy hole-light hole splitting	42.8 meV	32.3 meV
Single-dot PL on-time fraction	95.80%	~ 95 %
Biexciton quantum yield*	7.3 % – 37.7 % (depending on the core radius)	~34 % (one example)
Spectral diffusion (standard deviation of PL peak energy)	0.65 meV	0.97 meV
PL Peak	460 nm - 640 nm	515 nm - 616 nm

*Biexciton quantum yields are calculated from following equation; $QY_{XX} = QY_X \times g^2(0)$.

Following the reviewer's suggestion, we have revised the introduction part of manuscript to clarify the aim and novelty of present work.

Change made in the manuscript

(page3, line 36-64)

Colloidal quantum dots (QDs) are free-standing nano-emitters that radiate photons of size-dependent adjustable wavelength with a narrow emission linewidth^{1,2}. Core-shell heterostructuring in these nanometer scale semiconductors allows one to tailor their photophysical characteristics³⁻¹⁰. Heteroepitaxy involves structural deformation of both core and shell materials in a way that the structural stress at the interface is efficiently alleviated. The structural deformation of core materials often appears to be enough to modify the electronic structure and photophysical characteristics¹¹⁻¹³. Specifically, asymmetric lattice strain imposed on QDs lifts the dark exciton states above the bright excitons and splits the bright excitons into two distinct states^{12,14,15}, accompanying the accelerated radiative decay, narrowed spectral linewidths and the reduced optical gain threshold which are crucial for QD applications in displays¹⁶⁻¹⁸, lasers¹⁹⁻²¹ and single-photon sources²².

The degeneracy lift in colloidal QDs was first reported from a spherical core-asymmetrically grown shell geometry¹⁹ and has been found even in a conventional spherical core-shell geometry^{16-18,21-24}. Despite progress, the relation between the structural factors and photophysical/electronic characteristics of

results QDs remains unknown. The structural complexity of QD heterostructures, which includes an asymmetric geometry¹⁹ of shell or variations in composition^{16-18,20-23} or crystal structure²⁴ of shell along with misfit defects of near the interface, is the main culprit that impedes systematic investigation on the degeneracy lift in QDs. This calls for a materials platform that reaches degeneracy lift in core-shell heterostructure of a tailored geometry without misfit defect formation.

In the present study, we devise a coherent pseudomorphic growth technique for ZnSe shell over CdSe core to reach degeneracy lift in a conventional core-shell geometry. Resulting CdSe-ZnSe QDs bear compositionally abrupt but strain-graded core-shell interface, in which the compressive strain imposed in CdSe is greatly intensified while the formation of misfit defects is effectively suppressed. We perform comprehensive study across structural analysis, ensemble and single-dot spectroscopic analyses and computational calculations on the strain-graded QDs of variable core-shell dimensions, and relate the structural factors and their photophysical properties. Finally, we reveal the potential of coherent pseudomorphic growth of centered core-shell geometry to expand the emission energy envelope of strain-graded QDs, covering a wider spectral range and opening new avenues for applications in photonics.

Comment #2-2: Asymmetric strain results in the splitting of exciton fine structure as heavy hole and light hole. I wonder if the significant difference of two emission states causes different stability. Do the linewidths, emission wavelengths, efficiency and energy difference between the split levels maintain during EL device operation?

Response: It seems that Reviewer #2 questions about the controllability of emission from heavy hole and light hole excitonic states using electric field, which could promise their usage as qubits in quantum information/computer applications. To answer the question, one needs to construct single-dot electroluminescence devices and characteristics them under varying electric fields (carrier pumping rates), which is far beyond the scope of present work.

Instead, we have tested EL characteristics of QD films upon operation. Firstly, we compare the changes in EL spectra before and after device operation (**Fig. R2-1a**). Secondly, we monitor the changes in EL spectra as a function of applied voltages (current densities). In this condition, QDs emit lights under an electric field and heat. (**Fig. R2-1b**). Finally, we monitor the changes in PL emission of QDs upon reverse voltage (QDs under electric field, no heat) (**Fig. R2-1c**).

Firstly, no significant difference is seen in EL spectra before and after device operation (**Fig. R2-1a**), indicating that excitonic states do not change upon operation. Secondly, we observe gradual shift of EL emission to lower energy regime along with spectral broadening as the applied voltage (or current density) increases (**Fig. R2-1b**). We observe similar trend, the red-shift and broadening in PL of QDs under electric field (**Fig. R2-1c**). These experiments show that EL emission dominantly comes from the heavy hole exciton state, and the EL emission moves to lower energy and becomes broader. The red-shift and spectra broadening are largely due to the stark effect. The disparity in EL spectra in **Fig. R2-1b** and PL spectra in **Fig. R2-1c** is attributed to the heat generated during the device operation. The stark effect and thermal broadening are commonly seen in other QD-LEDs to different extents depending on the structure of QDs. We note, due to the significant energy shift and broadening of EL spectra (up to 63 meV for 8 V), we could not assess the independent characteristics of heavy hole and light hole emission states.

Fig. R2-1 | (a) EL spectra before and after device operation, (b) EL spectra upon applied voltages (current densities), (c) PL emission of QDs upon reverse voltage (under electric field).

Comment #2-3: The authors reconfirmed that the increased exciton energy is due to strain, not interface alloying using chemical etching. This part suddenly came up and was an unkind part to readers. I recommend to add more comments to the paper or the supplementary information.

Response: The increase in exciton energy of CdSe core upon ZnSe shell growth can be attributed to either the decrease in effective exciton radius due to the compression of CdSe core or the increase in the bulk bandgap of the core as a result of Zn diffusion into CdSe core to form CdZnSe alloy. These two suggestions are discernable in respect to the reversibility upon ZnSe shell etching. Specifically, the former, compression of CdSe core, is reversible upon ZnSe shell etching, whereas the latter, the diffusion of Zn into CdSe core to form CdZnSe, is irreversible upon ZnSe shell etching. From the etching experiment, we monitor that the photophysical characteristics return to their original states, indicating that the increased exciton energy in CdSe-ZnSe sg-QDs upon ZnSe shell growth is indeed attributed to the compressive strain given to CdSe core rather than the Zn diffusion into CdSe core (**Supplementary Fig. 7**).

To aid readers' understanding, we have elaborated the caption for **Supplementary Fig. 7** as below.

Change made in Supplementary Information

(Supplementary Fig. 7)

Supplementary Fig. 7 | (a) TEM images, (b) PL spectrum, (c) changes in the peak PL energy, (d) PL decay dynamics and (e) radiative recombination time upon chemical etching of CdSe ($r = 2.5 \text{ nm}$)-ZnSe ($H = 4.0 \text{ nm}$) QDs. Scale bars in (a) are 20 nm. It is noted that upon the shell etching, the peak PL energy, decay dynamics and exciton lifetimes return to the characteristics of original CdSe core-only QDs ($r = 2.5 \text{ nm}$). The complete recovery of photophysical characteristics upon ZnSe shell etching indicate that the blue shift in PL and the acceleration of exciton lifetime seen in CdSe-ZnSe sg-QDs upon ZnSe growth are indeed

attributed to the compressive strain imposed on CdSe core by ZnSe shell, rather than the irreversible compositional change in the core, for example, the inter-diffusion of Zn into CdSe to form CdZnSe core.

Comment #2-4: In Supplementary Note 1, the caption needs to be corrected from (a) and (b) to (a) and (c).

Response: We thank Reviewer #2 for pointing out a typo in the **Supplementary Note 1**. We have fixed the typo in the revised version of Supplementary Information.

Change made in Supplementary Information

(Supplementary Note 1)

Atomic structure of CdSe (111) surfaces: In (a) and (c), surfaces are 100% Cd and Se-terminated, respectively. In (b) and (d), a Cd and a Se vacancy for every four surface atoms are generated, respectively. The sites of the vacancies are denoted by a dotted circle. Solid lines represent the cell boundary. Only the atoms at the surface are shown for clarity.

Comment #2-5: I recommend the change of graphs in Supplementary Fig 13a and 13b from CdSe/ZnSe to CdZnSe core/shell QDs. Although the spectra of CdSe/ZnSe QDs appeared in the front part of the paper, CdZnSe core/shell QDs only appeared in Fig 4. Therefore, it would be better to show the graphs of CdZnSe core/shell QDs.

Response: As Reivwer #2 suggested, we have changed the data in **Supplementary Fig. 14a,b** to Cd_{0.25}Zn_{0.75}Se ($r = 1.8 \text{ nm}$)-ZnSe ($H = 5.0 \text{ nm}$) sg-QDs.

Change made in Supplementary Information

(Supplementary Fig. 14)

Supplementary Fig. 14 | (a) PL emission spectrum (grey dot) and its component, $1S_{HX,em}$ (red shading) and $1S_{LX,em}$ (blue shading), of a single Cd_{0.25}Zn_{0.75}Se ($r = 1.8 \text{ nm}$)-ZnSe ($H = 5.0 \text{ nm}$) sg-QD. Emission linewidth for PL spectrum and the energy splitting are noted. (b) Histogram showing the energy split ($1S_{HX,em} - 1S_{LX,em}$) gained from 38 individual Cd_{0.25}Zn_{0.75}Se ($r = 1.8 \text{ nm}$)-ZnSe ($H = 5.0 \text{ nm}$) sg-QDs. The mean split energy is ca. 28.0 meV. (c) $1S_{HX,em} - 1S_{LX,em}$ energy splits for the CdSe ($1.4 \text{ nm} \leq r \leq 4.5 \text{ nm}$)-ZnSe ($H = 5.0 \text{ nm}$) sg-QDs and Cd_{0.25}Zn_{0.75}Se ($1.8 \text{ nm} \leq r \leq 2.4 \text{ nm}$)-ZnSe ($H = 5.0 \text{ nm}$) sg-QDs.

Reviewer #3 (Publish after revision noted):

General Comment: The authors have developed defect-free core-shell Quantum Dots (QDs) that exhibit spectrally pure, stable, and polarized emission. This was achieved by introducing asymmetric compressive strain within the core region, which lifts the degeneracy of exciton states. Specifically, the authors synthesized CdSe-ZnSe QDs, reducing compositional and structural inhomogeneity by using a Se-rich CdSe core. This approach avoids unintentional alloying at the interface and reduces spectral broadening. As a result, the authors have attained near-unity photoluminescence quantum yields (PL QYs), narrow spectral linewidths, and suppressed spectral diffusion at the individual dot level, and high spectral purity in ensemble. Although some of finding has been published in the author's previous paper with Klimov, there are some interesting and new points here that are of interests to the readers. Especially, they demonstrated the feasibility of extending their approach to the entire visible spectrum. The reported approach can be helpful for a variety of photonic applications based on QDs, such as LED, laser, and single photon sources. Overall, the analysis and the experimental data support the author's hypothesis. I believe this work can be published after addressing the following concerns.

Response: We sincerely appreciate Reviewer #3's positive evaluation (Publish after major revision noted) and constructive comments and suggestions on our work. All authors thoroughly went over all comments and suggestions, prepared the responses, and revised the manuscript accordingly. Specifically, per Reviewer 3's comments, we provide the comparison of our sg-QDs to others having either intentional or unintentional alloyed interfaces, DFT calculation results to explain the changes in the magnitude of structural anisotropy in CdSe core upon ZnSe shell growth, and HR-TEM and XRD analysis to confirm the crystallographic structure of our QDs. Below, we enclose the point-by-point responses to each comment raised by the Reviewer #3. We look forward that the revised version of manuscript sufficiently addresses all the issues and now suitable for publication in *Nature Communications*.

Comment #3-1: Asymmetrically stressed QDs with similar compositions but cg-shells heterostructure have been reported (ref 27). The novelty of the authors' work lies in the synthesis of a compositionally abrupt interface, which reduces the spectral broadening originating compositional and structural inhomogeneity. This pertains to the ensemble level. While the authors only present a comparison of the FWHM of individual QD in Fig. 1k, please provide ensemble-level comparisons. Additionally, the authors mention that compositional gradients reduce the magnitude of compressive stress, whereas abrupt compositional interfaces enhance the strain imposed in CdSe core. Please attach a comparison of these two QDs to substantiate this point.

Response: Our experiments show that the compressive strain given in CdSe by ZnSe shell brings about the degeneracy lift, leading to the acceleration of radiative recombination rate and the spectral linewidth narrowing. In addition, the compressive strain suppresses the exciton-phonon coupling, which further reduces the spectral linewidth. The extent of compressive strain given in CdSe core is exploited by adopting a compositionally abrupt interface, which is translated into the heavy-hole light-hole energy split (ΔE_{split}), the acceleration of radiative recombination time (τ_X) and narrowing of PL spectral linewidth (FWHM). We compare these characteristics of our CdSe-ZnSe sg-QDs grown from Se-rich CdSe core with CdSe-ZnSe sg-QDs having an unintentional alloying interface grown from Cd-rich CdSe core (**Fig. R3-1**) and compositional gradients (**Table R3-1**). Specifically, per Reviewer 3's request, we include the ensemble level comparison. Our CdSe-ZnSe sg-QDs with an abrupt interface have the largest energy split (ΔE_{split}), the narrowest PL spectral linewidth in both single-dot and ensemble levels, and the shortest radiative recombination time. We include the comparison in the revised version of Supplementary Information (**Supplementary Fig. 1, Table 3**).

Fig. R3-2 | (a) Inductively coupled plasma atomic emission spectroscopy (ICP-AES) elemental analysis showing the chemical compositions of Cd and Se in Cd-terminated CdSe and Se-terminated CdSe of the same radius (r) of 2.5 nm. (b) EDS elemental mapping for Cd-terminated CdSe-ZnSe sg-QDs. Scale bar is 10 nm. (c) Absorption, (d) PL spectra, (e) radiative single exciton recombination times (τ_X) and (f) PL linewidth of ensemble CdSe ($r = 2.5 \text{ nm}$)-ZnSe sg-QDs with Cd-rich CdSe core upon ZnSe growth ($0 \leq H \leq 5 \text{ nm}$). Optical characteristics of CdSe-ZnSe sg-QDs with Se-rich CdSe core ($r = 2.5 \text{ nm}$) upon ZnSe shell growth (pink circles) are shown for comparison. (g) Heavy hole-light hole energy split (ΔE_{split}) measured from single-dot PL spectrum of 60 individual CdSe ($r = 2.5 \text{ nm}$)-ZnSe ($H = 5.0 \text{ nm}$) QDs with Cd-rich CdSe core (the mean split = 35.7 meV, upper panel) *versus* Se-rich CdSe core (the mean split = 42.8 meV, lower panel) are shown.

Table R3-1 | Comparison of photophysical characteristics of strain-graded CdSe-ZnSe QDs and compositionally-graded CdSe-Cd_xZn_{1-x}Se QDs¹

Key characteristics	Strain-graded QDs (this study)	Compositionally-graded QDs (ref. 1)
Structure	CdSe-ZnSe QDs with an abrupt interface	CdSe-Cd _x Zn _{1-x} Se QDs with composition gradient
PL QY	up to ~ 100 %	up to ~85 %
Ensemble PL linewidth	45 meV	68.6 meV
Single-dot PL linewidth	19.1 meV (in average)	24.1 meV (in average)
Radiative decay rate	0.092 ns ⁻¹	0.063 ns ⁻¹
Heavy hole-light hole splitting	42.8 meV	32.3 meV
Single-dot PL on-time fraction	95.80%	~ 95 %
Biexciton quantum yield*	7.3 % – 37.7 % (depending on the core radius)	~34 % (one example)
Spectral diffusion (standard deviation of PL peak energy)	0.65 meV	0.97 meV
PL Peak	460 nm - 640 nm	515 nm - 616 nm

*Biexciton quantum yields are calculated from following equation; $QY_{XX} = QY_X \times g^2(0)$.

Comment #3-2: It is noted that as the shell thickness increases, the effective compressive strain also increases, while the heavy hole-light hole energy split exhibits a decreasing trend after reaching its peak. The authors attribute this phenomenon to the growth of a bulk-like ZnSe shell. However, this explanation may not be readily understandable. In biaxially strained CdSe-CdS core-shell QDs reported by Fan et al., the strain is provided by asymmetric shell, when the shell grows thick, it is hard to avoid grow shell on the thin side of QD. However, the mechanism in ZnSe based strained dots is different. Why the splitting also saturates after certain shell thickness? Further analysis or supplementation with additional DFT theoretical simulations would be beneficial to clarify this aspect.

Response: We thank Reviewer #3 for his/her constructive comment that indeed helps enrich scientific discussion in our work. Our findings highlight two primary components of strain induced by the shell on the core: effective compressive strain and strain asymmetry. Through calculations and spectroscopic data, we demonstrated that the effective compressive strain increases with the thickness of the shell. As observed, the heavy hole-light hole energy split, which is indicative of strain asymmetry, initially increases until it reaches a peak, and then decreases. This asymmetry arises from the lattice mismatch between the core and shell in CdSe-ZnSe QDs, where the mismatch varies along the crystal's (*A*, *B*) and *C* axes, leading to directional differences in strain levels.

To identify the origin of the asymmetry, we performed Density Functional Theory (DFT) calculations for CdSe/ZnSe interface. Our interface models have the same number of CdSe layers (12), while the number of ZnSe layers is increased from 0 to 24, as illustrated in **Fig. R3-2a**. As we chose the interface normal direction to be $[10\bar{1}0]$ (*A* axis), we were able to see the biaxial compressive strain along $[0001]$ (*C* axis) and $[1\bar{2}10]$ direction, which is perpendicular to the *A* and *C* axes. As shown in **Fig. R3-2b**, the lattice contracts along both directions because of the smaller lattice constants of ZnSe, but the degree of the compressive strain shows differences. Initially, the strain along the $[1\bar{2}10]$ direction changes more rapidly compared to the *C* axis, creating strain asymmetry. However, as the shell grows further, the difference between these directional strains diminishes, as shown in **Fig. R3-2c**. This behavior is in good agreement with the experimentally observed increase and subsequent decrease in strain asymmetry, while the overall lattice is still more compressively strained. These findings and corresponding calculations have been added to the manuscript and supplementary information for further clarification.

Fig. R3-2 | (a) An atomistic model of CdSe/ZnSe interface, (b) The degree of biaxial strain along [0001] and $[1\bar{2}10]$ directions, (c) Strain anisotropy, which is calculated by the ratio between the strain along the two directions.

Change made in the manuscript

(Page 6, line 112-113)

However, compressive strain in CdSe core develops differently depending on the crystal axes. Specifically, in CdSe (radius, $r = 2.5 \text{ nm}$)-ZnSe (shell thickness, $H = 5.0 \text{ nm}$) heterostructure, the mean compressive strain (β , $\Delta d/d \times 100$ (%)) of entire CdSe core regime along [0002] and $[11\bar{2}0]$ are measured to be -3.08 % and -4.33 %, respectively (**Fig. 1c-e** and **Supplementary Fig. 5**). **This asymmetric strain between the basal direction, $[11\bar{2}0]$, and its orthogonal direction, [0002] is in good agreement with our DFT calculation results (Supplementary Note 3).**

(Page 8, line 149-152)

This would explain the tendency – the magnitude of energy splitting increases along the growth of ZnSe epilayers to reach the peak ($\Delta E_{split} = 55 - 60 \text{ meV}$ at $H = 2.0 - 3.0 \text{ nm}$) and decreases down to 50 meV when bulk like ZnSe shell ($H \geq 4.0 \text{ nm}$) is grown (**Fig. 2d**). **We attribute this intriguing behavior to the nonlinear change of the asymmetric strain for the thickness of the ZnSe shell (Supplementary Note 3), although the exact thickness might be different due to the simplified geometry in our calculation.**

Change made in Supplementary Information

(Supplementary Note 3)

Supplementary Note 3. Asymmetric strain of CdSe/ZnSe interface

(a) An atomistic model of CdSe/ZnSe interface, (b) The degree of biaxial strain along $[0001]$ and $[1\bar{2}10]$ directions, (c) Strain anisotropy, which is calculated by the ratio between the strain along the two directions.

The biaxial compressive strain of CdSe by ZnSe shell was modeled by performing DFT calculations. In our CdSe/ZnSe interface models, the number of ZnSe layers is increased from 0 to 24, while the number of CdSe layers is kept to 12. Since the interface normal direction is along $[10\bar{1}0]$ (*A axis*), we observe the biaxial compressive strain along $[0001]$ (*C axis*) and $[1\bar{2}10]$ direction, which is perpendicular to the *A* and *C* axes. The lattice contracts along both directions because of the smaller lattice constants of ZnSe. Initially, the strain along the $[1\bar{2}10]$ direction changes more rapidly compared to the *C* axis, creating strain asymmetry. However, the difference between these directional strains diminishes as the ZnSe layers are added. This computational finding is in good agreement with the observed increase and subsequent decrease in strain asymmetry, while the overall lattice is still more compressively strained.

Comment #3-3: The wurtzite/zinc blende polytypic structure has been considered as another possible reason to HH-LH splitting (Nat. Photon. 18, 186–191 (2024)). Could the authors please check their samples and identify whether this could another reason for the observed splitting? XRD and HRTEM will be helpful.

Response: As Reviewer #3 pointed out, the wurtzite/zincblende polytypic structure can be a possible reason for HH-LH splitting. In this case, the lattice stacking along the $[11\bar{2}0]_{wz}$ zone axis can be the criteria to identify the structural change: if the structure is purely wurtzite, it is stacked in ABABAB along $[0002]$ direction (**Fig. R3-3a**), while the structure is polytypic, the stacking manner changes from ABABAB to ABCABC in the shell phase (**Fig. R3-3b**).

In our CdSe-ZnSe sg-QDs, we observe ABABAB stacking along the $[0002]$ direction (**Fig. R3-3c,d**), supporting that the heterostructures indeed consist of wurtzite CdSe core and wurtzite ZnSe shell. We also confirm wurtzite characteristic peaks from XRD patterns of our CdSe-ZnSe sg-QDs (**Fig. R3-4**). From the experimental results, we exclude the wurtzite/zincblende polytypic structure as the reason to explain HH-LH splitting.

Fig. R3-3 | Schematics showing (a) wurtzite along the $[11\bar{2}0]_{wz}$ zone axis versus (b) zincblende along the $[110]_{zb}$ zone axis. (c,d) HR-TEM image along the $[11\bar{2}0]_{wz}$ zone axis of our CdSe-ZnSe sg-QD. AB-stacking of lattice along $[0002]$ is highlighted in (d) for visual clarity.

Fig. R3-4 | XRD patterns of powder samples of (wz) CdSe-(wz) ZnSe sg-QDs. The bulk wz ZnSe patterns are shown for comparison.

References

- 1 Park, Y. S., Lim, J. & Klimov, V. I. Asymmetrically strained quantum dots with non-fluctuating single-dot emission spectra and subthermal room-temperature linewidths. *Nat. Mater.* **18**, 249-255 (2019).
- 2 Held, J. T. *et al.* Obtaining Structural Parameters from STEM–EDX Maps of Core/Shell Nanocrystals for Optoelectronics. *ACS Applied Nano Materials* **1**, 989-996 (2018).
- 3 Huang, L. *et al.* Synthesis of Colloidal Quantum Dots with an Ultranarrow Photoluminescence Peak. *Chemistry of Materials* **33**, 1799-1810 (2021).
- 4 Zhang, J., Li, C., Li, J. & Peng, X. Synthesis of CdSe/ZnSe Core/Shell and CdSe/ZnSe/ZnS Core/Shell/Shell Nanocrystals: Surface-Ligand Strain and CdSe–ZnSe Lattice Strain. *Chemistry of Materials* **35**, 7049-7059 (2023).
- 5 Efros, A. L. *et al.* Band-edge exciton in quantum dots of semiconductors with a degenerate valence band: Dark and bright exciton states. *Physical Review B* **54**, 4843-4856 (1996).
- 6 Chang, J. H. *et al.* Impact of Morphological Inhomogeneity on Excitonic States in Highly Mismatched Alloy ZnSe_{1–x}Te_x Nanocrystals. *The Journal of Physical Chemistry Letters* **13**, 11464-11472 (2022).
- 7 Jang, E. *et al.* *Effect of Tellurium Doping on Optoelectronic Properties of Blue ZnTeSe Quantum Dots* (Research Square Platform LLC, 2022).
- 8 Imran, M. *et al.* Molecular Additive-Assisted Tellurium Homogenization in ZnSeTe Quantum Dots. *Advanced Materials* **n/a**, 2303528 (2023).

Reviewer #1 (Remarks to the Author):

The authors' efforts in this new version have substantially improved the work. My recommendation is to consider the work to be published in Nature Communications after addressing the following points:

- 1) The authors reported that the bi-exciton quantum yield depends on the core size of the QDs. Can the authors quantitatively compare the bi-exciton yield with the core radius of the QDs and discuss this comparison in the main text or SI?
- 2) Can the authors comment on the theoretically expected bi-exciton quantum yield, considering a radiative decay rate of 0.092 ns^{-1} , and compare it with their experimental data?
- 3) The growth of a zinc blende ZnSe shell on a ZB CdSe ($r = 2.5 \text{ nm}$) core with a Se-rich surface, and the comparison of photophysical characteristics with those of a zinc blende CdSe core with a Cd-rich surface, is very interesting. However, the structural data provided is insufficient to confirm the ZB cores and subsequent ZB shell growth. The authors should perform XRD to properly identify the ZB structure of the QDs and discuss this data in the SI with a brief comment on the comparison in the main text between ZB and WZ.
- 4) Why is the separation of excitonic states into two distinct peaks in the absorption and PL spectra less evident in the case of zinc blend core-shell QDs?
- 5) The authors used TOPSe and DDPSe in the synthesis of the QDs. What is the role of DDPSe in the synthesis?
- 6) ODE is known to polymerize high-temperature synthesis of QDs. Can the author comment on this?
- 7) It is not clear from the description in the methods section whether oleate-capped QDs were used in devices or if ligand exchanges were performed. please clarify.

Reviewer #2 (Remarks to the Author):

I recommend the acceptance of this article as the authors resolved or explained all the issues that reviewers were concerned about.

Reviewer #3 (Remarks to the Author):

The authors have made significant improvements and addressed my comments well. I think the manuscript can be accepted.

The following are some minor issues:

1. The interplanar spacing is about 0.375 nm in Fig. R3-3, greater than 0.345 nm in Fig. 1. The HRTEM in Fig. R3-3 looks like along the $[0002]$ rather than the $[112(_)0]$. However, the QDs exhibit a high-purity wurtzite structure, as shown in the XRD pattern, so the wurtzite/zincblende polytypic structure can be excluded as a reason to explain the HH-LH splitting.
2. The length of all the scale bars in Fig. 1 is missing.

Reviewer #1 (Publish after minor revision noted):

Comment #1-1: The authors' efforts in this new version have substantially improve the work. My recommendation is to consider the work to be published in in Nature Communications after addressing the following points. The authors reported that the bi-exciton quantum yield depends on the core size of the QDs. Can the authors quantitatively compare the bi-exciton yield with the core radius of the QDs and discuss this comparison in the main text or SI?

Response: The second-order correlation function at time 0 of individual QDs denotes the ratio between biexciton quantum yield and single exciton quantum yield ($g^{(2)}(0) = QY_{XX}/QY_X$). We provide the second-order correlation function graphs for CdSe-ZnSe sg-QDs of varying core radii ($1.5 \text{ nm} \leq r \leq 4.5 \text{ nm}$) with thick ZnSe shell ($H = 5.0 \text{ nm}$) (**Fig. R1**). We also include them in the revised version of Supporting Information (**Supplementary Fig. 10**) and reference in the revised manuscript.

Fig. R1 | The second-order correlation function graphs of CdSe-ZnSe ($H = 5.0 \text{ nm}$) sg-QDs with different core radii $r =$ (a) 1.5 nm, (b) 2 nm, (c) 3 nm, (d) 3.5 nm, (e) 4 nm and (f) 4.5 nm. The values of $g^{(2)}(0)$ are noted in the figure.

Change made in the manuscript

(page8, line 170)

The type I band alignment with the smooth potential profile across CdSe core and ZnSe shell in sg-QDs facilitates funneling of charge carriers¹ into CdSe core, as reflected by the suppression of fluorescence intermittency (blinking) under the continued photo-irradiation (**Fig. 3a**). In addition, the smooth potential profile at the interface aids to suppress non-radiative Auger recombination processes² (**Fig. 3b** and **Supplementary Fig. 10**).

Change made in Supplementary Information

(Supplementary Fig. 10)

Supplementary Fig. 10 | The second-order correlation function graph of CdSe-ZnSe ($H = 5.0$ nm) sg-QDs with different core radii $r =$ (a) 1.5 nm, (b) 2 nm, (c) 3 nm, (d) 3.5 nm, (e) 4 nm and (f) 4.5 nm. The values of $g^{(2)}(0)$ are noted in the figure.

Comment #1-2: Can the authors comment on the theoretically expected bi-exciton quantum yield, considering a radiative decay rate of 0.092 ns^{-1} , and compare it with their experimental data?

Response: Reviewer #1 suggested calculating the theoretical bi-exciton quantum yield. However, the bi-exciton quantum yield cannot be solely explained by the radiative decay rate. It can be experimentally acquired either from the second-order correlation function measurement or the pump-fluence dependent TCSPC measurement. From the pump fluence PL decay dynamics of CdSe ($r = 2.5 \text{ nm}$)-ZnSe ($H = 5.0 \text{ nm}$) in an ensemble solution (**Fig. R2**), we could assess the bi-exciton decay rate ($k_{XX} = 2 \text{ ns}^{-1}$) and single-exciton decay rate ($k_X = 0.092 \text{ ns}^{-1}$), implying a bi-exciton quantum yield (18 %) after consideration of the statistical scaling and Auger decay. This is indeed in good agreement with the bi-exciton quantum yield (16 %) gained from the second-order correlation function measurement.

Fig. R2 | PL decay dynamics of CdSe ($r = 2.5 \text{ nm}$)-ZnSe ($H = 5.0 \text{ nm}$) sg-QDs composed with single exciton (X) or single exciton and bi-exciton (X+XX) from low or high pump fluence. Fitting of them gives single exciton and bi-exciton lifetimes.

Comment #1-3: The growth of a zinc blende ZnSe shell on a ZB CdSe ($r = 2.5$ nm) core with a Se-rich surface, and the comparison of photophysical characteristics with those of a zinc blende CdSe core with a Cd-rich surface, is very interesting. However, the structural data provided is insufficient to confirm the ZB cores and subsequent ZB shell growth. The authors should perform XRD to properly identify the ZB structure of the QDs and discuss this data in the SI with a brief comment on the comparison in the main text between ZB and WZ.

Response: As Reviewer #1 recommended, we performed the XRD measurement on zincblende *versus* wurtzite CdSe-ZnSe QDs (**Fig. R3**). Despite the reviewer shows his/her interest, the zincblende-wurtzite comparison is irrelevant to the main storyline of the manuscript, and we are afraid that this could distract the readers. Therefore, we decide not to include it in the MS or SI.

Fig. R3 | XRD patterns of (a) zincblende CdSe core ($r = 2.5$ nm) and CdSe ($r = 2.5$ nm)-ZnSe ($H = 3.0$ nm) and (b) wurtzite CdSe core ($r = 2.5$ nm) and CdSe ($r = 2.5$ nm)-ZnSe ($H = 3.0$ nm). Characteristics peaks for bulk zincblende (wurtzite) CdSe and ZnSe are noted at the bottom and top, respectively.

Comment #1-4: Why is the separation of excitonic states into two distinct peaks in the absorption and PL spectra less evident in the case of zinc blend core-shell QDs?

Response: The wurtzite CdSe has an intrinsic energy separation between HH-LH states (25 meV), which is absent in zincblende CdSe. This energy split is enlarged to be discerned in the absorption and PL spectra when the asymmetric strain is given upon ZnSe shell growth. The hydrostatic strain given in zincblende CdSe does not result in the HH-LH split. This contrast yields the disparity observed in the separation of excitonic states between zincblende and wurtzite CdSe-ZnSe QDs.

Comment #1-5: The authors used TOPSe and DDPSe in the synthesis of the QDs. What is the role of DDPSe in the synthesis?

Response: Reviewer #1 wonders the role of DPPSe that is used as Se precursor in the synthesis of CdZnSe alloyed QDs. To achieve homogeneous CdZnSe QDs, one needs to mitigate the reactivity difference between cation precursors (*i.e.*, Cd(OA)₂, Zn(OA)₂) and anion (Se) precursor. DPPSe, the secondary phosphine-based selenium precursor, serves this purpose by exhibiting high reactivity with metal precursors, effectively reducing them into metal-Se complex. The use of DPPSe at high reaction temperature alleviates the reactivity difference between cation precursors (Cd(OA)₂, Zn(OA)₂) and anion precursor (DPPSe) allowing to gain homogeneously alloyed CdZnSe QDs.

Comment #1-6: ODE is known to polymerize high-temperature synthesis of QDs. Can the author comment on this?

Response: As Reviewer #1 points out, 1-octadecene (ODE) is known to polymerize into poly-ODE at high temperature¹. We note that careful Schlenk line technique (i.e., chemicals were carefully degassed under vacuum for 4 hours and the chemistry is conducted in an inert atmosphere) aids to minimize the formation of poly-ODE. In addition, centrifugation (>6,000 rpm) at room temperature can remove poly-ODE from the QD products. The successful elimination of poly-ODE is evidenced by the electrical properties of QD-LEDs, which otherwise show the low current density and the delayed turn-on voltage.

Comment #1-7: It is not clear from the description in the methods section whether oleate-capped QDs were used in devices or if ligand exchanges were performed. please clarify.

Response: We used oleate-capped QDs for the photonic applications. In response to Reviewer #1's comment, we add the description of QD preparation for the photonic application.

Change made in the manuscript

(page17, line 357-358)

Device Fabrications. For QD-LEDs fabrication (ITO/Zn_{0.9}Mg_{0.1}O/QDs/CBP/MoO_x/Al), 20 mg/mL Zn_{0.9}Mg_{0.1}O nanoparticles were spun-cast on the ITO substrate at 4000 rpm for 30 s and annealed at 75 °C for 30 min in a glovebox. 14 mg/mL QDs were spun-cast at 4000 rpm for 30 s on ZnO/ITO and annealed at 75 °C for 30 min in the glovebox. CBP (60 nm), MoO_x (10 nm), and Al (120 nm) were thermally deposited on the QDs/ZnO/ITO films under a pressure of $\sim 10^{-6}$ Torr at a deposition rate of 1.0–1.5, 0.1–0.2, and 1.0–2.0 Å/s, respectively. QDs were used after purification without further ligand exchange process. The devices were encapsulated for subsequent characterization.

Reviewer #3 (Publish after minor revision noted):

The authors have made significant improvements and addressed my comments well. I think the manuscript can be accepted.

The following are some minor issues:

Comment #3-1: The interplanar spacing is about 0.375 nm in Fig. R3-3, greater than 0.345 nm in Fig. 1. The HRTEM in Fig. R3-3 looks like along the [0002] rather than the [112($\bar{1}$)0]. However, the QDs exhibit a high-purity wurtzite structure, as shown in the XRD pattern, so the wurtzite/zincblende polytypic structure can be excluded as a reason to explain the HH-LH splitting.

Response: We thank Reviewer #3 for his/her comment on our mistake for the zone axis direction. Upon reevaluation, we have confirmed that the reviewer's observation is indeed correct.

Comment #3-2: The length of all the scale bars in Fig. 1 is missing.

Response: The scale bar is not noted in the figure but mentioned in the caption for visual clarity.

References

- 1 Dhaene, E., Billet, J., Bennett, E., Van Driessche, I. & De Roo, J. The Trouble with ODE: Polymerization during Nanocrystal Synthesis. *Nano Letters* **19**, 7411-7417 (2019).

Reviewer #1 (Remarks to the Author):

The authors have addressed the final comments. I recommend the manuscript for publication in Nat. Comms.

Reviewer #1 (Publish as is):

General Comment: The authors have addressed the final comments. I recommend the manuscript for publication in Nat. Comms.

Response: We sincerely appreciate Reviewer #1's positive evaluation on our work, Reviewer #1's comments in the revision process have been invaluable in improving the overall quality and integrity of our manuscript and enhancing the impact of our research.